# DFlow: Learning to Synthesize Better Optical Flow Datasets via a Differentiable Pipeline

**Kwon Byung-Ki**[1]**, Nam Hyeon-Woo**[1]**, Ji-Yun Kim**[1]**, Tae-Hyun Oh**[1,2,3]
[1]Department of Electrical Engineering, POSTECH    [2]Graduate School of AI, POSTECH
[3]Institute for Convergence Research and Education in Advanced Technology, Yonsei University
{byungki.kwon, hyeonw.nam, junekim, taehyun}@postech.ac.kr

## Abstract

Comprehensive studies of synthetic optical flow datasets have attempted to reveal what properties lead to accuracy improvement in learning-based optical flow estimation. However, manually identifying and verifying the properties that contribute to accurate optical flow estimation require large-scale trial-and-error experiments with iteratively generating whole synthetic datasets and training on them, *i.e.*, impractical. To address this challenge, we propose a differentiable optical flow data generation pipeline and a loss function to drive the pipeline, called DFlow. DFlow efficiently synthesizes a dataset effective for a target domain without the need for cumbersome try-and-errors. This favorable property is achieved by proposing an efficient dataset comparison method that uses neural networks to approximately encode each dataset and compares the proxy networks instead of explicitly comparing datasets in a pairwise way. Our experiments show the competitive performance of our DFlow against the prior arts in pre-training. Furthermore, compared to competing datasets, DFlow achieves the best fine-tuning performance on the Sintel public benchmark with RAFT.

## 1 Introduction

Optical flow is a fundamental computer vision problem to find dense pixel-wise correspondences between two subsequent frames in a video. Optical flow is indeed a key building block in many practical applications, including video understanding, action analysis, video enhancement, editing, 3D vision, *etc*. Recently, optical flow has been significantly advanced by learning-based approaches with deep neural networks (Fischer et al., 2015; Ilg et al., 2017; Ranjan & Black, 2017; Hui et al., 2018; Sun et al., 2018; Teed & Deng, 2020) in terms of accuracy and efficiency.

A driving force of these prior arts is large-scale supervised datasets. However, it is difficult to collect a reasonable amount of real-world optical flow labels. Thus, they exploited large-scale synthetic datasets, *e.g.*, Fischer et al. (2015); Mayer et al. (2016), which has become the standard in optical flow, *e.g.*, training on FlyingChairs (Fischer et al., 2015) followed by FlyingThings3D (Mayer et al., 2016). After the seminal studies, there have been various efforts to build different synthetic datasets (Gaidon et al., 2016; Richter et al., 2017; Lv et al., 2018; Oh et al., 2018; Aleotti et al., 2021). Despite the vast efforts of these studies, it remains unclear which factors are important for an effective synthetic dataset construction against the target domain.

Instead of manually identifying important design criteria, AutoFlow (Sun et al., 2021) pioneers the first learning-based approach to go beyond being heuristic by posing data generation as a hyperparameter optimization problem maximizing validation performance on a target dataset. AutoFlow generates data samples by composing simple 2D layers with non-differentiable hyperparameters, which are optimized by sampling-based evolutionary search. The use of evolutionary search requires large resources, which is burdensome because each target scenario requires to re-generate different datasets.

To address this challenge, we propose DFlow, which is an efficient synthetic optical flow dataset generation method. We compose each data sample by simple differentiable graphic operations, such as warping layer and real-world effects, so that each sample can be parameterized in a learnable manner. This allows us to exploit efficient gradient descent methods to generate each sample, and

thereby DFlow is more than an order of magnitude efficient than AutoFlow in GPU hours when constructing the same amount of training data.

We also introduce a new loss function that learns the data parameters by contrasting a target dataset from a base dataset, *e.g.*, FlyingChairs. Since directly using large datasets in the contrastive learning process is cumbersome, we approximate the base and target datasets by two neural networks trained on respective datasets as proxies. This approximation allows an end-to-end differentiable pipeline from the data parameters to the loss function.

Through comprehensive experiments, we show that DFlow is effective in both pre-training and fine-tuning. The DFlow data has a size of $512 \times 384$, which is the same as FlyingChairs, but the RAFT network (Teed & Deng, 2020) pre-trained on DFlow achieves comparable performance compared to the high-resolution competing datasets (Sun et al., 2021; Mayer et al., 2016). In addition, compared to competing datasets, the RAFT model initially pre-trained on DFlow achieves the best fine-tuning performance on the Sintel public benchmark. We summarize our contributions as follows:

- A simple and efficient differentiable data generation pipeline for optical flow (refer to Table 1);

- A contrastive-style learning scheme and its loss function by approximating expensive dataset-to-dataset comparison by leveraging proxy neural networks (refer to Sec 3);

- Compared to competing datasets, DFlow achieves the best fine-tuning performance on the Sintel public benchmark with RAFT. (refer to Table 4).

## 2 RELATED WORK

**Optical Flow.** Dense optical flow estimation is to find pixel-wise correspondences from the brightness patterns of images (Gibson, 1950; Gibson & Carmichael, 1966; Horn & Schunck, 1981). After conventional optimization algorithms (Black & Anandan, 1993; Zach et al., 2007), deep-learning algorithms (Fischer et al., 2015; Ilg et al., 2017) become dominant due to their computational efficiency and reasonable performance. Prior arts (Xu et al., 2017; Bailer et al., 2017; Wulff et al., 2017; Sun et al., 2018) have attempted to implement explicit neural modules that are suitable for optical flow estimation. Recently, RAFT (Teed & Deng, 2020) adopts recurrent architectures and achieves a notable performance improvement, which is represented as state-of-the-art.

Those recent advances in learning-based approaches require large-scale data with ground-truth, but labeling dense optical flow is a highly undetermined task, *i.e.*, challenging (Fischer et al., 2015). The previous real-world datasets have been built under sophisticated labeling conditions, including the special sensor hardware, controlled environment, or limited objects (Scharstein & Szeliski, 2002; Scharstein & Pal, 2007; Geiger et al., 2012; Kondermann et al., 2014). It leads to the limitation of the size of datasets. To relieve this issue, synthetic datasets (Fischer et al., 2015; Mayer et al., 2016) have been proposed and achieved remarkable accuracy despite the gap between real and synthetic datasets. After that, previous arts endeavor to construct more realistic synthetic datasets (Gaidon et al., 2016; Richter et al., 2017; Lv et al., 2018). The prior arts (Aleotti et al., 2021; Han et al., 2022) generate the subsequent frames and ground-truth optical flow by warping the previous frame. These do not handle the photometric inconsistency that is common in real-world scenes. In this work, we propose a differentiable synthetic data generation pipeline with the target knowledge so that the generated dataset could improve its performance more.

**Learning-based Optical Flow Dataset.** AutoFlow (Sun et al., 2021) is the first learning-based data generation approach in optical flow, but with the sampling-based evolutionary search for non-differentiable optimization. It is our closest related work in the sense that they learn the data generation parameters for performance improvement on the specific target dataset. However, distinctively, our method is the first differentiable method to learn data generation parameters, which leads to a more efficient pipeline than AutoFlow in terms of computation cost and GPU hours for data generation. We list other differences in Table 1. Recently, RealFlow (Han et al., 2022) proposes an iterative learning framework that learns enhanced flow estimation and pseudo ground-truth generation, alternatively. Different from our work, this work suggests a framework including iterative model training and dataset generation, and does not suggest a goodness measure of a resulting dataset, which we address. Other than optical flow, there are also interesting attempts to generate synthetic data in learnable

|  | AutoFlow (Sun et al., 2021) | DFlow (Ours) |
|---|---|---|
| Learning method | Evolutionary search | Gradient descent |
| Computational resource | 48 P100 GPUs | A single V100 GPU |
| GPU hours for constructing a dataset | 336 days / P100 | 9.3 days / V100 |
| Data resolution | $1280 \times 720$ | $512 \times 384$ |

Table 1: **Comparison of dataset generation methods of AutoFlow and DFlow (ours).**

ways (Sixt et al., 2018; Yang & Deng, 2020; Kaspar et al., 2019; Zhao et al., 2021) for other computer vision applications.

## 3 DIFFERENTIABLE DATA GENERATION PIPELINE

Our data generation pipeline synthesizes subsequent frames $\mathbf{I}_t$ and $\mathbf{I}_{t+1}$, and a flow (motion) map $\mathbf{F}_{GT}$ between those two frames. We concisely denote a pair of subsequent frames and a flow map as a data sample. We parameterize each data sample with a learnable parameter vector $\boldsymbol{\theta}$ that characterizes a composition of elemental graphic operations, such as color perturbation and geometric warping. We refer to the composition process as a data generator, which is designed to be differentiable.

With this differentiable data generator part, we specifically seek to synthesize optical flow data improving the accuracy on the target dataset. We define a loss function $\mathcal{L}_{\text{total}} = \mathcal{L}_{\text{task}} + \mathcal{L}_{\text{reg}}$ to effectively update the learnable parameters, which drives the data generation to an improving direction. With this loss, we efficiently learn the parameters $\{\boldsymbol{\theta}\}$ in a fully-differentiable way.

The overview of our data generation pipeline is illustrated in Fig. 1, called DFlow. In Sec. 3.1, we first describe the design of the loss function. In Sec. 3.2, we describe the data generator method from the data parameters $\{\boldsymbol{\theta}\}$. Further detailed descriptions and implementation details can be found in Appendix A.

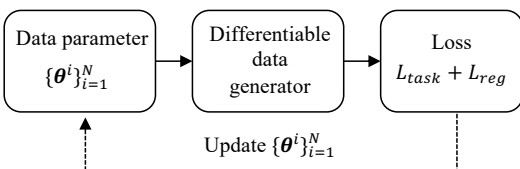

Figure 1: **An overview of DFlow.** We parameterize data samples to be synthesized. The learnable data parameters $\{\boldsymbol{\theta}\}$ are updated by our proposed loss function.

### 3.1 TASK LOSS FUNCTION

Given our differentiable data generator, a sample parameter $\boldsymbol{\theta}$ is rendered to a data sample, *i.e.*, a pair of subsequent frames $\mathbf{I}_t(\boldsymbol{\theta})$ and $\mathbf{I}_{t+1}(\boldsymbol{\theta})$ and those corresponding flow map $\mathbf{F}_{GT}(\boldsymbol{\theta})$. By rendering a collection of data samples with the data generator and a set of $\{\boldsymbol{\theta}\}$, we can construct a dataset. To find an updated parameter $\boldsymbol{\theta}^*$ that improves the accuracy of the optical flow model, we need an effective criterion to drive the parameter update.

As a tractable criterion to efficiently find such a dataset, we are motivated by learning-to-augment and dataset condensation approaches (Sixt et al., 2018; Kaspar et al., 2019; Zhao et al., 2021). They synthesize data to have characteristics similar to target data under the motivation that learning with two similar datasets is likely to yield accuracy improvement (Ben-David et al., 2010; Kaspar et al., 2019), which is typically achieved by distribution matching. However, directly comparing combinatorial pairs of samples in the target dataset and synthesized one is computationally expensive, and even intractable during iteratively updating a synthesized dataset.

To efficiently deal with this issue, we propose to encode the target dataset into a proxy neural network, called target network. With an optical flow neural network $f_{\text{target}}(\mathbf{I}_t, \mathbf{I}_{t+1}) \rightarrow \mathbf{F}$, we pre-train the network with a given small target optical flow dataset, which yields the trained target network. We deem the target network as a differentiable proxy of the target dataset that encodes the knowledge of target data. Compared to using the target data as a set in set-to-set comparison, leveraging this differentiable proxy allows the whole procedure to be tractable. With the target network, we define a target loss $\mathcal{L}_{\text{target}}\left(f_{\text{target}}\left(\mathbf{I}_t(\boldsymbol{\theta}), \mathbf{I}_{t+1}(\boldsymbol{\theta})\right), \mathbf{F}_{GT}(\boldsymbol{\theta})\right)$ that measures flow errors for a data sample associated with $\boldsymbol{\theta}$ in the view of the target domain. We use the same loss used for pre-training the target network, *e.g.*, the sequence loss for RAFT (Teed & Deng, 2020), for the target loss $\mathcal{L}_{\text{target}}(\cdot)$.

With the target network and loss, we could first attempt to update the parameters $\{\boldsymbol{\theta}\}$ by

$$\{\boldsymbol{\theta}^*\} = \arg\min_{\{\boldsymbol{\theta}\}} \sum_i \mathcal{L}_{\text{target}}\left(f_{\text{target}}\left(\mathbf{I}_t(\boldsymbol{\theta}_i), \mathbf{I}_{t+1}(\boldsymbol{\theta}_i)\right), \mathbf{F}_{GT}(\boldsymbol{\theta}_i)\right). \quad (1)$$

If the target loss is small for a data sample, it means that the target network is familiar with the given data sample, and the sample is close to one of the target data, *i.e.*, similar characteristics to the target data. In this sense, Eq. (1) distills the target knowledge into generated data. However, in our empirical preliminary study, we observed instability and under-fitting issues with Eq. (1), where optimizing the loss is stuck at a high value. We assume that target networks trained on the popular benchmarks (Butler et al., 2012; Menze & Geiger, 2015) may not produce sufficiently rich training signals for synthesizing data, which might be due to limited scales of those real-world benchmarks.

To complement more informative training signals, we employ the contrastive-style learning scheme by using another comparator network, called base network, which is trained with a common large-scale synthetic dataset, *e.g.*, (Fischer et al., 2015; Mayer et al., 2016), as a base dataset. Similar to the target network, we pre-train the base network but on a large-scale dataset, FlyingChairs (Fischer et al., 2015), which is randomly generated. We use the base network similarly to the target network except for maximizing flow errors $\mathcal{L}_{\text{base}}$ to implement a contrastive behavior, so that a resulting data sample should be closer to the target dataset while being different from the base (common) dataset. This contrastive behavior can generate combinatorial diversity of gradient signals by pairwise contrasting, which may more strongly drive the data generation to an improving direction in favor of a target domain than using the target network alone. To implement the contrastive behavior, we define a wrapping loss function, $\mathcal{L}_{\text{task}}(\mathcal{L}_{\text{target}}, \mathcal{L}_{\text{base}}): \mathbb{R}^2 \rightarrow \mathbb{R}$, called task loss.

**What Function Type is Suitable for $\mathcal{L}_{\text{task}}$ .** There are many candidates for $\mathcal{L}_{\text{task}}$ satisfying the aforementioned criteria. As a function form, we are first motivated by sample-wise weighting distillation (Zhou et al., 2021), where the loss function is dynamically weighted according to each data sample. The loss function is weighted by the target and base losses in a multiplication form. We additionally examine addition forms to select the best one. We consider tanh, sigmoid, and exponential functions to implement bounded contrasting behaviors.[1] Table 2 shows the accuracy comparison of different forms of loss functions using $1,000$ data samples generated with the Sintel dataset (Butler et al., 2012) as a target. The addition form with the exponential function shows promising results in terms of performance, which corresponds to

| Form | $\mathcal{L}_{\text{task}}$ | Dataset | |
|---|---|---|---|
| | | Sintel clean | Sintel final |
| Multiplication | Exponential | 4.26 | 4.49 |
| | Sigmoid | 4.12 | 4.35 |
| | Tanh | 2.02 | 3.35 |
| Addition | Exponential | **1.85** | **3.09** |
| | Sigmoid | 2.00 | 3.13 |
| | Tanh | 1.95 | 3.21 |

*Bold denotes the best.

Table 2: **AEPE of different forms of $\mathcal{L}_{\text{task}}$.** We train the RAFT network with different loss forms. The addition form with exponential shows promising results. Refer to Appendix A.2 for details.

$$\mathcal{L}_{\text{task}}(\mathcal{L}_{\text{target}}, \mathcal{L}_{\text{base}}) = \mathcal{L}_{\text{target}} + \alpha \exp(-\beta \tfrac{\mathcal{L}_{\text{base}}}{\mathcal{L}_{\text{target}}+\epsilon} + \gamma), \quad (2)$$

where we set the balance parameters $\{\beta, \gamma\}$ to $\{1, 0\}$, and especially set $\alpha$ to 20 for the experiments in Table 2. We found that, depending on $\alpha$, generated data characteristics are distinctive. When generating our final dataset, we first generate subsets of data with different $\alpha$ values, and ensemble the subsets into a single dataset by taking union, denoted as $\{\alpha\}$ *combination*. This is found to be notably effective. More details of the total loss can be found in Appendix.

## 3.2 DATA GENERATOR

In this section, we describe the differentiable data generator that synthesizes a data sample from a parameter $\boldsymbol{\theta}$. As shown in Fig. 2, the data generator is composed of geometric warping, flow field translation, layer composition, color perturbation, and real-world effects. The data generator takes $N$ pairs of layer images and masks, $\{\mathbf{L}_0^l\}_{l=1}^N$ and $\{\mathbf{M}_0^l\}_{l=1}^N$. Similar to Oh et al. (2018), We randomly sample the layer images from public image datasets (Kuznetsova et al., 2020; Perazzi et al., 2016; Mayer et al., 2016; Lin et al., 2014) and layer masks from a segmentation dataset (Everingham et al.,

---

[1]We observe that contrasting without bounds may lead to instability and divergence. We list the specific equations of the used loss functions in Appendix A.2.

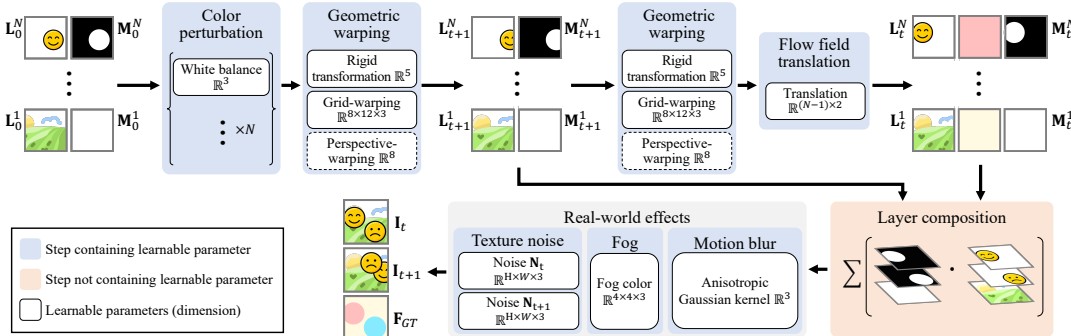

Figure 2: **Schematic of data generator.** The data generator is composed of color perturbation, two steps of geometric warpings, flow field translation, layer composition, and real-world effects. The whole pipeline is differentiably parameterized.

---

**Algorithm 1:** PyTorch-style pseudo-code for data generator.

```
# W1, W2:  geometric warping parameters
# Ds:  translation parameters
# C: white balance parameters
# R: Real-world effect parameters
# images0, masks0:  initial layer images and masks

# Applying color perturbation
images0 = ColorPerturbation(images0, C)

# Warping the images and masks
images1, masks1 = GeometricWarping(images0, masks0, W1)
W2s = FlowFieldTranslation(W2, Ds)
images2, masks2 = GeometricWarping(images1, masks1, W2s)

# Superimposing the images, masks, and flow fields
image1 = LayerComposition(images1, masks1)
image2 = LayerComposition(images2, masks2)
flow = LayerComposition(W2s, masks1)

# Applying real-world effects to the subsequent images
image1, image2 = RealworldEffect(image1, image2, R)
```

---

2010). The data generator then outputs synthesized subsequent frames $\mathbf{I}_t$ and $\mathbf{I}_{t+1}$ with ground-truth optical flow $\mathbf{F}_{GT}$. All the following pipeline is parameterized by $\boldsymbol{\theta}$.

**Geometric Warping.** Geometric warping consists of rigid transformation, perspective warping, and grid warping. The geometric warping generates a warping field. In the first geometric warping, we apply the same warping field $\mathbf{W}_{0 \to t+1}$ to $\{\mathbf{L}_0^l\}_{l=1}^N$ and $\{\mathbf{M}_0^l\}_{l=1}^N$ and generate the layer images and masks at the frame $t+1$, $\{\mathbf{L}_{t+1}^l\}_{l=1}^N$ and $\{\mathbf{M}_{t+1}^l\}_{l=1}^N$. In addition to this globally shared geometric warping, we model the local movement of each layer, *i.e.*, segmented objects, in the second geometric warping step. To generate complex optical flows of the frame $t$, we can apply independent warping fields to each layer image and mask. However, we observe poor optimization behaviors when we use all independent warping fields on each layer. Thus, we propose to use decomposed warping parameters for time $t$ to reduce the number of parameters. We use a anchor geometric warping, $\mathbf{W}_{t+1 \to t}$, shared across all layers and flow field translation $(\Delta x, \Delta y)$, $\{\mathbf{D}^l \in \mathbb{R}^2\}_{l=1}^N$, for each layer, and construct each warping of layers by $\mathbf{W}_{t+1 \to t}^l = (\mathbf{D}^l \circ \mathbf{W}_{t+1 \to t})$, where $\circ$ denotes the warping operation. This strategy is also beneficial in terms of optimization stability. Finally, the operations of geometric warping and flow field translation are as follows:

$$\mathbf{L}_{t+1}^l = \mathbf{W}_{0 \to t+1} \circ \mathbf{L}_0^l, \quad \mathbf{L}_t^l = \mathbf{W}_{t+1 \to t}^l \circ \mathbf{L}_{t+1}^l = (\mathbf{D}^l \circ \mathbf{W}_{t+1 \to t}) \circ \mathbf{L}_{t+1}^l \tag{3}$$

$$\mathbf{M}_{t+1}^l = \mathbf{W}_{0 \to t+1} \circ \mathbf{M}_0^l, \quad \mathbf{M}_t^l = \mathbf{W}_{t+1 \to t}^l \circ \mathbf{M}_{t+1}^l = (\mathbf{D}^l \circ \mathbf{W}_{t+1 \to t}) \circ \mathbf{M}_{t+1}^l. \tag{4}$$

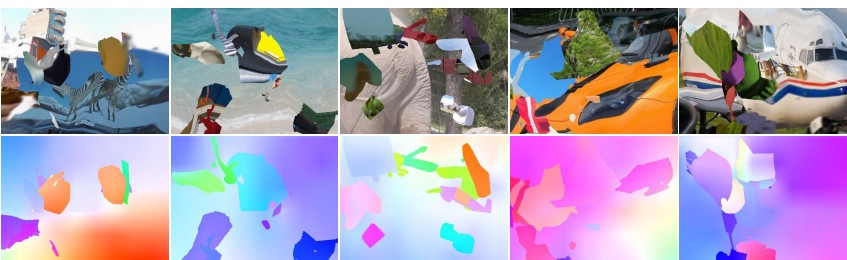

Figure 3: **Generated data samples.** (top) current frames; (bottom) optical flow visualizations.

**Layer Composition.** After the geometric warping, we have $N$ pairs of layer images, masks, and warping fields in order of depth. We superimpose each stack of layers to generate subsequent frames $\mathbf{I}_t$ and $\mathbf{I}_{t+1}$, with ground-truth optical flow $\mathbf{F}_{GT}$. We can leverage alpha blending and softmax splatting (Niklaus & Liu, 2020). Both strategies show comparable performance, but we mainly use the softmax splatting strategy in our experiments. The comparison of alpha blending and softmax splatting can be found in Sec. 4.

**Color Perturbation and Real-world Effects.** Prior studies (Mayer et al., 2018; Sun et al., 2021) have shown that synthetic data containing real-world effects, such as texture noises, fog, and motion blur, often brings the performance improvement of optical flow networks in generalization. Inspired by the observations, we introduce color perturbation and real-world effects into our data generator as well. Color perturbation adjusts the white balance of each layer image, and our real-world effects apply texture noises, fog, and motion blur, which are all parameterized to be controlled and updated.

We synthesize optical flow data by applying the above components: $\{\mathbf{L}_0^l\}_{l=1}^N, \{\mathbf{M}_0^l\}_{l=1}^N \rightarrow \{\mathbf{I}_t, \mathbf{I}_{t+1}, \mathbf{F}_{GT}\}$. Note that we apply regularizations to the grid warping and texture noise, *i.e.*, grid and noise regularizations. The detail of each component, including regularizations, can be found in Appendix A.3-A.6. The overall differentiable data generator pipeline is summarized in Algorithm 1 as a pseudo-code. Each component has its own parameters, which are updated by our task loss and regularizations. See generated samples in Fig. 3.

**Summarization.** We provide the summarization of DFlow below:

1. We train base and target networks on base and target datasets, respectively.
2. We fix both networks and update the data parameter $\boldsymbol{\theta}$ using our loss function.
3. We train optical flow networks with the generated dataset.

## 4 RESULTS

In this section, we analyze the effects of our method in pre-training and fine-tuning perspectives with different optical flow models. We report the average end-point error (AEPE) for Sintel (Butler et al., 2012) and AEPE & F1 for KITTI 2015 (Menze & Geiger, 2015). From this section, we refer to our dataset generated by targeting Sintel with RAFT proxy models as DFlow, unless specified otherwise. Other details of experiment setups and implementation can be found in Appendix B.

**Pre-training RAFT Results.** Pre-training performance is one of the key factors in evaluating the applicability of optical flow datasets (Fischer et al., 2015; Mayer et al., 2016; 2018; Aleotti et al., 2021; Sun et al., 2021) by testing on benchmarks, *i.e.*, Sintel and KITTI 2015 datasets. We train the RAFT networks (Teed & Deng, 2020) from scratch on respective competing datasets, including our DFlow. Table 3-(a) shows the pre-training results of each dataset. Compared to FlyingChairs which has the same data resolution as DFlow, the model trained on DFlow outperforms the one trained on FlyingChairs in both Sintel and KITTI 2015 datasets. Despite the fact that DFlow is around $\frac{1}{4}$ resolution of AutoFlow, DFlow achieves the best performance on the KITTI 2015 and Sintel clean datasets. We postulate that the real-world textures used in DFlow led to performance improvement on KITTI 2015. Also, we use both Sintel clean and final as the target datasets to generate DFlow, unlike AutoFlow using Sintel final only. It might affect performance on the Sintel clean and final datasets. Figure 4 shows the qualitative results obtained from FlyingChairs and DFlow. These results show that DFlow is effective for learning an accurate model in challenging scenes, such as shaded, foggy, and motion-blurred scenes.

Table 3: **Pre-training results.** We train the (a) RAFT (Teed & Deng, 2020) and (b) FlowNet (Fischer et al., 2015) (c) GMA (Jiang et al., 2021) networks on respective pre-training datasets, and evaluate on the evaluation datasets, *i.e.*, Sintel and KITTI 2015. Chairs→Things denotes the heterogeneous dataset experiment pre-training on FlyingChairs followed by FlyingThings3D.

| Model | Pre-training dataset | Evaluation dataset | | |
| --- | --- | --- | --- | --- |
| | | Sintel clean | Sintel final | KITTI 2015 |
| | | AEPE | AEPE | AEPE / F1 |
| (a) RAFT | FlyingChairs | 2.28 | 4.51 | 9.85 / 37.56 |
| | AutoFlow | 2.08 | **2.75** | 4.66 / - |
| | DFlow (Ours) | **1.81** | 2.93 | **4.59 / 15.03** |
| | Chairs→Things | 1.43 | 2.71 | 5.04 / 17.40 |
| (b) FlowNet | FlyingChairs | 4.71 | 6.22 | 20.36 / 62.20 |
| | AutoFlow | 5.43 | 6.03 | 19.64 / 43.95 |
| | DFlow (Ours) | **3.45** | **4.73** | **12.94 / 38.95** |
| (c) GMA | Chairs→Things | - | - | 4.69 / 17.10 |
| | DFlow-GMA (Ours) | - | - | **4.47 / 13.10** |

*\***Bold** denotes the best.

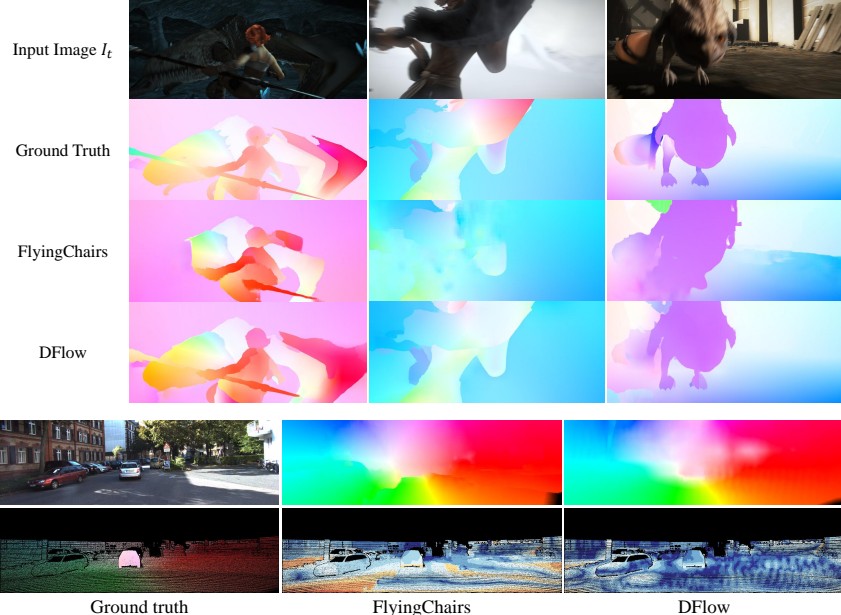

Figure 4: **Qualitative results.** Top: pre-training results on the Sintel final pass. Bottom: pre-training results on KITTI 2015. The RAFT network pre-trained on DFlow shows robust results in challenging shaded, foggy, and motion-blurred scenes.

**Model Generalization Property.** Since DFlow is generated with RAFT proxy networks, a question naturally arises about the effectiveness of the generated dataset and data generation pipeline on other model architectures. To evaluate the generalization ability of the generated dataset, *i.e.*, DFlow, we pre-train the Flownet network on the DFlow dataset generated with the RAFT network. As shown in Table 3-(b), FlowNet pre-trained on DFlow shows noticeable improvement over the ones pre-trained on the competing datasets. We further investigate the generalization ability of the data generation pipeline. In this experiment, we select the KITTI 2015 dataset as the target dataset and generate the DFlow-GMA dataset using the GMA network (Jiang et al., 2021). Using our DFlow-GMA dataset, we train the GMA network and list the performance on KITTI 2015 in Table 3-(c). The GMA network trained on our DFlow-GMA achieves higher performance than the one trained on FlyingChairs followed by FlyingThings3D. From these observations, we postulate that our generated data and generation pipeline are agnostic to the deep network structure.

Table 4: **Fine-tuning results on public benchmarks.** We report F1 scores for KITTI 2015 and AEPE for Sintel public benchmarks. C, K, S, T, and H denote FlyingChairs, KITTI 2015, Sintel, FlyingThings3D, and HD1K (Kondermann et al., 2014). The results of Sintel public benchmarks are evaluated without the *warm-start* initialization (Teed & Deng, 2020).

| Dataset schedule | Sintel clean | Sintel final | KITTI 2015 |
|---|---|---|---|
| C → T → TSKH/K (Teed & Deng, 2020) | 1.94 | 3.18 | 5.10 |
| AutoFlow → TSKHV (Sun et al., 2021) | 2.01 | 3.14 | **4.78** |
| DFlow (Ours) → TSKH/K | **1.62** | **3.07** | 5.03 |

*\*__Bold__ denotes the best.*

Table 5: **Analysis of data generation components.** We analyze the effects of each generation component. The experiments are conducted by adding or removing one of the generation components. The default setting is with all components and the number of foregrounds parameter in the range of 8 to 12 except the $\{\alpha\}$ combination, which is annotated with underlines.

| Experiments | State | Evaluation dataset | | |
|---|---|---|---|---|
| | | Sintel clean | Sintel final | KITTI 2015 |
| | | AEPE | AEPE | AEPE / F1 |
| Data update | On | 1.86 | 3.04 | 5.21 / 16.27 |
| | Off | 9.70 | 9.67 | 17.31 / 46.84 |
| $\{\alpha\}$ combination | Off | 1.86 | 3.04 | 5.21 / 16.27 |
| | On | 1.90 | 3.02 | 4.90 / 15.78 |
| Layer composition | softmax splatting | 1.86 | 3.04 | 5.21 / 16.27 |
| | alpha blending | 1.95 | 2.83 | 5.23 / 16.06 |
| Number of foregrounds | 0 | 4.10 | 4.89 | 7.56 / 23.76 |
| | 2 | 2.15 | 3.32 | 5.73 / 17.60 |
| | 4 | 2.00 | 3.15 | 5.41 / 16.33 |
| | 8-12 | 1.86 | 3.04 | 5.21 / 16.27 |
| Color perturbation | On | 1.86 | 3.04 | 5.21 / 16.27 |
| | Off | 2.07 | 3.20 | 4.87 / 16.09 |
| Motion blur | On | 1.86 | 3.04 | 5.21 / 16.27 |
| | Off | 1.95 | 3.57 | 5.06 / 16.24 |
| Fog | On | 1.86 | 3.04 | 5.21 / 16.27 |
| | Off | 1.98 | 3.20 | 5.04 / 15.68 |
| Texture noise | On | 1.86 | 3.04 | 5.21 / 16.27 |
| | Off | 2.02 | 3.15 | 5.47 / 16.56 |
| Noise regularization | On | 1.86 | 3.04 | 5.21 / 16.27 |
| | Off | 1.91 | 3.09 | 4.97 / 15.87 |
| Grid regularization | On | 1.86 | 3.04 | 5.21 / 16.27 |
| | Off | 1.88 | 3.10 | 5.09 / 16.35 |
| Target dataset | Sintel | 1.86 | 3.04 | 5.21 / 16.27 |
| | KITTI 2015 | 2.14 | 3.79 | 4.31 / 14.29 |

**Fine-tuning Results on Public Benchmarks.** Table 4 shows the fine-tuning results on the public benchmark test sets with corresponding dataset schedules. DFlow improves the original RAFT recipe (C→T→TSKH/K) on both benchmarks by replacing the conventional initial dataset schedule, *i.e.*, FlyingChairs followed by FlyingThings3D (C→T), with our DFlow. In particular, compared to the competing datasets, DFlow achieves the best fine-tuning performance on the Sintel with RAFT. This result validates that DFlow as a pre-training dataset effectively affects fine-tuning results.

**Analysis of Components.** We analyze the effects of each generation pipeline component: $\{\alpha\}$ combination, the number of foregrounds, color perturbation, real-world effects, regularizations, and target dataset. For the fair and quick experiments, we generate 2k training data and train RAFT networks with the same training details. As shown in Table 5, we add or remove each of the components from a default setting to measure their effects with the pre-training experiment. The default setting indicates the dataset with all components and the number of foregrounds parameter in the range of 8 to 12 except the $\{\alpha\}$ combination. We observe that applying each component brings consistent performance improvement on the target dataset, *i.e.*, Sintel, which is used for generating

DFlow; whereas the results on KITTI 2015, which is not a target data of DFlow, are inconsistent with the results of Sintel.

- *Data update.* This analysis compares DFlow with randomly initialized data without any optimization, *i.e.*, randomly generated by our pipeline. We train and compare the RAFT networks with those datasets, which show the effectiveness gap between the two datasets.

- $\{\alpha\}$ *Combination.* We collect the same amount of data by combining subsets of data, which are obtained from diverse $\alpha$ values. This balances the performance between evaluation datasets because of the regularization effect.

- *Layer composition.* The softmax splatting and alpha blending show comparable performance on Sintel. The softmax splatting improves the performance in the Sintel clean pass, but shows the performance drop in the Sintel final pass.

- *Number of foregrounds.* Without any foreground, we observe the significant performance drop on Sintel and KITTI 2015. Adding only 2 foregrounds brings notable performance improvement.

- *Color perturbation and real-world effects.* Without the color perturbation, the performance on Sintel drops moderately. Removing the motion blur significantly affects the accuracy on Sintel, especially in the Sintel final pass. The effects of fog and texture noise are moderate.

- *Regularizations.* Removing the regularizations shows a slight performance drop in Sintel.

- *Target dataset.* The generated dataset optimized to KITTI 2015 shows a notable performance gain on KITTI 2015, while the performance on Sintel significantly drops. This may be caused by the distribution gap between the two datasets.

**Robustness to Corrupted Data (Whether Artifact).** The photometric inconsistency degrades the robustness of optical flow estimation. For example, the rainy scene occurs photometric inconsistency. We evaluate the performance of RAFT on rainy scenes of Virtual KITTI (Gaidon et al., 2016). We choose FlyingChairs and RealFlow (Han et al., 2022) as baselines. Using the rainy scenes of Virtual KITTI as a target dataset, we generate an additional dataset, DFlow-V. Note that the target of DFlow is the Sintel dataset. Table 6 lists the performance of rainy scenes of virtual KITTI. As Han et al. (2022) have mentioned their limitation of discontinuous illumination, the performance of RealFlow is lower than others. DFlow outperforms the baselines, and DFlow-

| Dataset | Rainy scene AEPE / F1 |
|---|---|
| FlyingChairs | 7.22 / 24.95 |
| RealFlow | 7.75 / 27.57 |
| DFlow | 5.87 / 22.89 |
| DFlow-V | **3.11 / 12.61** |

*\***Bold** note the best.

Table 6: **Analysis of photometric inconsistency scenario.**

V, which of the target is Virtual KITTI, shows much higher performance rather than others. From these results, we assume that our method can distill the characteristics of photometric inconsistency.

## 5    CONCLUSION AND DISCUSSION

We propose a new data generation pipeline for training optical flow networks. Our pipeline consists of geometric warping, real-world effects, etc., which are all parameterized differentiably. We propose a new objective function that drives our data optimization by leveraging the compressed knowledge of the proxy networks pre-trained on target and base datasets, respectively. Optical flow models trained on our datasets achieve favorable or superior performance against the competing datasets on pre-training and fine-tuning experiments. We conclude our paper with a discussion section.

**Discussion.** We use the pre-defined elementary data generation operations, *e.g.*, fog, geometric warping, *etc*. While DFlow shows effectiveness in the real-world dataset, *i.e.*, KITTI 2015, the pre-defined and restricted operations might not span all the real-world effects. Thus, diverse and complementary operations would further improve expressiveness and may lead to additional performance improvement. Our method also aims at a specific target dataset. For that, we need a target network trained on the target dataset, which requires at least some amount of optical flow annotations. To mitigate the requirement of the supervised data, it would be an interesting future direction to investigate the way to train the target network in an unsupervised method.

## ETHICS STATEMENT

This paper develops an optical flow data generation method and uses the public datasets (Kuznetsova et al., 2020; Perazzi et al., 2016; Mayer et al., 2016; Lin et al., 2014; Everingham et al., 2010) for data generation. Therefore, the author does not expect any potential ethical issues related to sensitive information of the dataset.

## REPRODUCIBILITY STATEMENT

We introduce the whole data generation pipeline in Sec. 3. In Sec. 3.1, we present the loss function for data optimization. In Sec. 3.2, we describe the data generator method from the parameters. We list more details of the loss function and data generation pipeline in Appendix A.

## ACKNOWLEDGEMENT

This work was partially supported by the National Research Foundation of Korea (NRF) grant (No. NRF-2021R1C1C1006799), and Institute of Information & communications Technology Planning & Evaluation (IITP) grant (No.2022-0-00124, Development of Artificial Intelligence Technology for Self-Improving Competency-Aware Learning Capabilities; No.2021-0-02068, Artificial Intelligence Innovation Hub) funded by the Korea government (MSIT). This project is the result of "HPC Support" Project supported by the "Ministry of Science and ICT" and NIPA.

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

## APPENDIX

We present implementation details and additional experimental results. The contents are listed as follows:

## CONTENTS

# A    DETAILS OF DIFFERENTIABLE DATA GENERATION PIPELINE

In this section, we describe the implementation details of the differentiable data generation pipeline.

## A.1    BASE AND TARGET NETWORKS

**Base Network.**  For the architecture, we use the same architecture for both base and target networks. We use the published weight of RAFT (Teed & Deng, 2020) as base network[2], which is pre-trained on FlyingChairs (Fischer et al., 2015) for 100k iterations with a batch size of 12, $496 \times 368$ image size, and learning rate $4 \times 10^{-4}$.

**Target Network.**  We fine-tune the base network on the Sintel datasets (Butler et al., 2012) for the target network. During fine-tuning, we only use the image crop as data augmentation and fine-tune the base network for 20k iterations with a batch size of 6, $768 \times 368$ image size, and learning rate $1.25 \times 10^{-4}$.

## A.2    THE FORM OF TASK LOSS FUNCTIONS

We introduce our task loss function $\mathcal{L}_{\text{task}}(\mathcal{L}_{\text{target}}, \mathcal{L}_{\text{base}})$ for contrastive-style learning and evaluate the candidate of the loss function in Sec. 3.1. Table 7 shows the used task loss functions in experiments of Table 2. We use the sequence loss of flow estimates of RAFT (Teed & Deng, 2020) for $\mathcal{L}_{\{target,base\}}$. Algorithm 2 is the pseudo-code for updating the parameters of each data sample.

Table 7: **The forms of task loss function.** We list the used forms of task loss function in Sec. 3.1. The form is the combination of {Multiplication, Addition} and {Exponential, Sigmoid, Tanh}. The task loss functions have the hyperparameters, $\alpha, \beta$, and $\gamma$.

| Task loss function $\mathcal{L}_{\text{task}}(\mathcal{L}_{\text{target}}, \mathcal{L}_{\text{base}})$ | |
| --- | --- |
| Multiplication & Exponential | $(1 - \alpha \exp(\beta\mathcal{L}_{\text{base}}/(\mathcal{L}_{\text{target}} + \epsilon) + \gamma))\mathcal{L}_{\text{target}}$ |
| Multiplication & Sigmoid | $(1 + \alpha\texttt{sigmoid}(\beta\mathcal{L}_{\text{target}}/(\mathcal{L}_{\text{base}} + \epsilon) + \gamma))\mathcal{L}_{\text{target}}$ |
| Multiplication & Tanh | $(1 + \alpha\texttt{tanh}(\beta\mathcal{L}_{\text{target}}/(\mathcal{L}_{\text{base}} + \epsilon) + \gamma))\mathcal{L}_{\text{target}}$ |
| Addition & Exponential | $\mathcal{L}_{\text{target}} + \alpha \exp(-\beta\mathcal{L}_{\text{base}}/(\mathcal{L}_{\text{target}} + \epsilon) + \gamma)$ |
| Addition & Sigmoid | $\mathcal{L}_{\text{target}} + \alpha\texttt{sigmoid}(\beta\mathcal{L}_{\text{target}}/(\mathcal{L}_{\text{base}} + \epsilon) + \gamma)$ |
| Addition & Tanh | $\mathcal{L}_{\text{target}} + \alpha\texttt{tanh}(\beta\mathcal{L}_{\text{target}}/(\mathcal{L}_{\text{base}} + \epsilon) + \gamma)$ |

## A.3    GEOMETRIC WARPING AND FLOW FIELD TRANSLATION.

Given $N$ pairs of layer images and masks, $\{\mathbf{L}_0^l\}_{l=1}^N$ and $\{\mathbf{M}_0^l\}_{l=1}^N$, we generate subsequent frames, $\mathbf{I}_t$ and $\mathbf{I}_{t+1}$, with ground-truth optical flow $\mathbf{F}_{GT}$,

$$\{\mathbf{L}_0^l\}_{l=1}^N, \{\mathbf{M}_0^l\}_{l=1}^N \rightarrow \{\mathbf{I}_t, \mathbf{I}_{t+1}, \mathbf{F}_{GT}\}. \tag{5}$$

We apply two steps of geometric warping with warping fields $\mathbf{W}_{0 \rightarrow t+1}$ and $\mathbf{W}_{t+1 \rightarrow t}$ which are computed from combinations of rigid transformation, perspective warping, and grid warping. First, we warp the layer images and masks, $\{\mathbf{L}_0^l\}_{l=1}^N$ and $\{\mathbf{M}_0^l\}_{l=1}^N$, to be the layer images $\{\mathbf{L}_{t+1}^l\}_{l=1}^N$ and masks $\{\mathbf{M}_{t+1}^l\}_{l=1}^N$ at the $t + 1$ frame by applying the same warping field $\mathbf{W}_{0 \rightarrow t+1}$ to $\{\mathbf{L}_0^l\}_{l=1}^N$ and $\{\mathbf{M}_0^l\}_{l=1}^N$.

$$\mathbf{L}_{t+1}^l = \mathbf{W}_{0 \rightarrow t+1} \circ \mathbf{L}_0^l, \quad \mathbf{M}_{t+1}^l = \mathbf{W}_{0 \rightarrow t+1} \circ \mathbf{M}_0^l, \tag{6}$$

where $\circ$ denotes the geometric warping operation according to a given warping field. In the next step, each layer image $\mathbf{L}_t^l$ and mask $\mathbf{M}_t^l$ at the $t$ frame is generated from the ones at the $t + 1$ frame, *i.e.*, $\mathbf{L}_{t+1}^l$ and $\mathbf{M}_{t+1}^l$. To simulate complex optical flows with minimal parameters, we introduce the flow field translation $\{\mathbf{D}^l\}_{l=1}^N$, which translates the warping field $\mathbf{W}_{t+1 \rightarrow t}$ and obtains each warping of layers: $\mathbf{W}_{t+1 \rightarrow t}^l = (\mathbf{D}^l \circ \mathbf{W}_{t+1 \rightarrow t})$. Using each warping of layers $\mathbf{W}_{t+1 \rightarrow t}^l$, we warp the layer images and masks at the $t + 1$ time.

$$\mathbf{L}_t^l = (\mathbf{D}^l \circ \mathbf{W}_{t+1 \rightarrow t}) \circ \mathbf{L}_{t+1}^l, \quad \mathbf{M}_t^l = (\mathbf{D}^l \circ \mathbf{W}_{t+1 \rightarrow t}) \circ \mathbf{M}_{t+1}^l. \tag{7}$$

---

[2]https://github.com/princeton-vl/RAFT

**Algorithm 2:** PyTorch-style pseudo-code for DFlow.

```
# G: data generator
# B: base network
# T: target network
# L: our task loss function
# Lt:  target loss function
# Lb:  base loss function
# θ:  data parameters
for l, m in loader:
    # Generate optical flow data
    # l:  layer images, m:  layer masks
    image1, image2, label = G(l, m, θ)

    # loss
    lt = Lt(B(image1, image2), label)
    lb = Lb(T(image1, image2), label)
    loss = L(lt, lb)

    # Update θ
    loss.backward()
    optimizer.step()
```

The above parameterization is efficient for approximating multi-layer geometric warpings because it only introduces two additional parameters per layer to generate the local movement of each layer from a single warping field $\mathbf{W}_{t+1 \to t}$. As aforementioned, we propose to use the two steps of warping with $\mathbf{W}_{0 \to t+1}$ and $\mathbf{W}_{t+1 \to t}$. This is distinctive from the existing works, including Fischer et al. (2015); Sun et al. (2021), where they only use a single step of warping from a randomly generated image to a synthetic next frame. This only parameterizes the next frames, not the reference frames randomly generated. This could not update the textures and mask shapes of the reference frames, whereas our method parameterizes both frames with two warping fields, $\mathbf{W}_{0 \to t+1}$ and $\mathbf{W}_{t+1 \to t}$. It allows us to jointly update subsequent frames.

### A.4  LAYER COMPOSITION

From the geometric warping, we have $N$ pairs of layer images, masks, and warping fields in order of depth. We adopt the softmax splatting (Niklaus & Liu, 2020) to compose the layer images at subsequent frames and warping fields with flow field translation, *i.e.*, $\mathbf{L}_t^l$, $\mathbf{L}_{t+1}^l$, and $\mathbf{W}_{t+1 \to t}^l$. We get the importance image $\boldsymbol{Z}$ and weight image $\boldsymbol{K}$ of each layer and frame,

$$\boldsymbol{Z}_t^l = \mathbf{M}_t^l a^l - \max_l(\mathbf{M}_t^l a^l), \tag{8}$$

$$\boldsymbol{K}_t^l = \frac{\exp(\boldsymbol{Z}_t^l)}{\sum_{l=1}^N \exp(\boldsymbol{Z}_t^l)}. \tag{9}$$

where $a^l = cl$ and $c$ is a constant value which we set 6 for layer composition. As the value of $c$ increases, sharper boundaries can be obtained. The subsequent frames and ground-truth optical flow are as follows:

$$\left\{ \sum_{l=1}^N \boldsymbol{K}_{t+1}^l \odot \mathbf{L}_{t+1}^l, \ \sum_{l=1}^N \boldsymbol{K}_t^l \odot \mathbf{L}_t^l, \ \sum_{l=1}^N \boldsymbol{K}_t^l \odot \mathbf{W}_{t+1 \to t}^l \right\} \ \to \{\mathbf{I}_{t+1}, \mathbf{I}_t, \mathbf{F}_{GT}\}, \tag{10}$$

### A.5  COLOR PERTURBATION AND REAL-WORLD EFFECTS

**Color Perturbation.** Color perturbation consists of $N$ white balances to adjust the color intensity of layer images. Each white balance has three-channel values from zero to one and has half of the chance to be applied. Color perturbation is the same for subsequent frames and optimized.

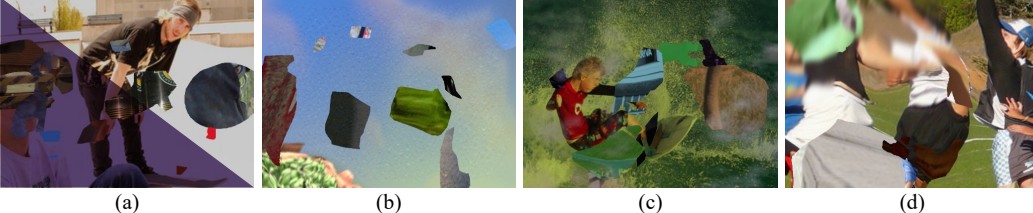

| (a) | (b) | (c) | (d) |

Figure 5: **Color perturbation and real-world effects.** Real-world effects are composed of texture noises, fog, and motion blur. From left to right: (a) color perturbation, (b) texture noises, (c) fog, (d) motion blur. (a) is an overlaid image before (upper right) and after color perturbation (lower left).

**Texture Noises.** We simulate the shot noise to mimic real-world photos. We apply three-channel texture noises to all of the pixels while masking the area where the noises are actually applied. Two different texture noises, $\mathbf{N}_t$ and $\mathbf{N}_{t+1}$, are applied to subsequent frames and optimized.

**Fog and Motion Blur.** Inspired by AutoFlow (Sun et al., 2021), we introduce two real-world effects; fog and motion blur. To generate a random fog, we follow the implementation of AutoFlow and superimpose the fog on $\mathbf{I}_t$ and $\mathbf{I}_{t+1}$. However, one difference is that we adjust the three-channel values of fog, which yields the colored fog. The fog is the same for the subsequent frames and is optimized as well. For the motion blur, we randomly sample object masks from PASCAL VOC (Everingham et al., 2010) and combine them to generate a motion blur mask. We use the 2D gaussian blur to approximate the motion blur kernel, and the motion blurred frames are alpha-blended to $\mathbf{I}_t$ and $\mathbf{I}_{t+1}$. The standard deviation of each axis and angle of rotation are parameters to be updated.

Figure 5 shows the color perturbation and real-world effects including texture noise, fog, and motion blur.

## A.6 REGULARIZATION LOSS

The grid warping and texture noise may take shortcuts only to minimize task loss. These behaviors are undesirable, and the generated data might not properly train the optical flow network. To handle this potential issue, we propose a regularization term $\mathcal{L}_{\text{reg}}$ consisting of grid and noise regularization losses, *i.e.*, $\mathcal{L}_{\text{reg}} = \mathcal{L}_{\text{grid}} + \mathcal{L}_{\text{noise}}$. The grid regularization $\mathcal{L}_{\text{grid}}$ prevents from producing infeasible 2D motion. We define the grid regularization loss $\mathcal{L}_{\text{grid}}$ as follows:

$$\mathcal{L}_{\text{grid}} = \max(0, \sum_{k=1}^{w}\sum_{j=1}^{h-1}[\mathbf{C}_t(k,j) - \mathbf{C}_t(k,j+1)] + \sum_{k=1}^{w-1}\sum_{j=1}^{h}[\mathbf{C}_t(k,j) - \mathbf{C}_t(k+1,j)]), \quad (11)$$

where $\mathbf{C}_t$ is the warped coordinate by the geometric warping $\mathbf{W}_{t+1\to t}$. $\mathcal{L}_{\text{grid}}$ gives a penalty when the warped coordinates of the previous grid exceed those of the next grid. The noise regularization $\mathcal{L}_{\text{noise}}$ prevents from producing too noisy images. We define the noise regularization loss $\mathcal{L}_{\text{noise}}$ as follows:

$$\mathcal{L}_{\text{noise}} = (\|\mathbf{N}_t\|_1 + \|\mathbf{N}_{t+1}\|_1), \quad (12)$$

where $\mathbf{N}_t$ and $\mathbf{N}_{t+1}$ are texture noises of the current and next frames, respectively. Finally, we obtain our total loss by adding the regularization loss to the task loss as:

$$\mathcal{L}_{\text{total}} = \mathcal{L}_{\text{task}}(\mathcal{L}_{\text{target}}, \mathcal{L}_{\text{base}}) + \mathcal{L}_{\text{reg}}. \quad (13)$$

With the total loss $\mathcal{L}_{\text{total}}$, we update the data parameters $\{\boldsymbol{\theta}\}$.

## A.7 DETAILS OF LEARNABLE PARAMETER $\boldsymbol{\theta}$

**Learning Rate of $\boldsymbol{\theta}$.** We set the learning rate of real-world effects as $3 \times 10^{-2}$. and the others as $\{1 \times 10^0, 1 \times 10^{-1}, 2 \times 10^{-2}\}$ depending on whether they are pixel-unit operations (*e.g.*, translation parameters) or not. We decay the learning rates linearly over the update iterations; the decay factor is $(1 - iteration/80)$, where $80$ is the maximum $iteration$ of updates. We distinguish the pixel-unit operations and the others as:

Table 8: **Generation details of DFlow.** Data augmentation used in generating DFlow.

| Data augmentation | Color jitter | | | | Random resize and crop | | |
|---|---|---|---|---|---|---|---|
| | brightness | contrast | saturation | hue | min scale | max scale | image crop |
| | 0.1 | 0.1 | 0.1 | 0.04 | 0.93 | 2.30 | $496 \times 368$ |

- Pixel-unit operation (learning rate of $1 \times 10^0$: translation of the rigid transformation, grid warping, perspective warping, and flow field translation
- Non-pixel-unit operation (learning rates of $\{1 \times 10^{-1}, 2 \times 10^{-2}\}$): rotation and scaling of the rigid transformation

**Initialization of $\theta$.** Following the implementation of AutoFlow (Sun et al., 2021), we sample the initial data parameters $\theta$ from the uniform distributions ranging from $a$ to $b$, *i.e.*, $[a, b]$.

- White balance: [0, 1]
- Rigid transformation (translation, rotation, scaling): ([-80, 80], [-5, 5], [0.75, 1.13])
- Grid warping: [0, 0]
- Perspective warping: [-25, 25]
- Flow field translation: [-50, 50]
- Texture noise: [-0.01, 0.01]
- Fog color: [0, 1]
- Motion blur (std of axis x, std of axis y, angle of rotation): ([1, 2], [3, 11], [0, 90])

## B  DETAILS OF EXPERIMENT SETUP

In this section, we present the generation details of the DFlow dataset. Then, we present the details of pre-training and fine-tuning results in Sec. 4 of the main paper.

### B.1  GENERATION DETAILS OF THE DFLOW DATASET

To generate the DFlow dataset, the data augmentation is included in the proposed differentiable data generation pipeline. We randomly augment each data sample before inputting them into the base and target networks. For stability of data generation, we stack 6 randomly augmented data samples and input them into networks. Each data sample of DFlow is updated up to 80 times and saved depending on the threshold 25 of the target network loss. The DFlow data has a size of $512 \times 384$, which is the same as FlyingChairs. Table 8 summarizes the data augmentation applied to each data sample.

### B.2  DETAILS OF PRE-TRAINING RESULTS

Except for the data augmentation, we set the training parameter settings by following the training details of RAFT (Teed & Deng, 2020) on FlyingChairs (Fischer et al., 2015), and pre-train the RAFT network on DFlow with a data size of 15k. We use the same data augmentation applied for generating the DFlow dataset. We early stop the training at 95k iteration.

### B.3  DETAILS OF MODEL GENERALIZATION PROPERTY RESULTS

For the FlowNet results, we use the Pytorch implementation of FlowNet[3]. We pre-train the FlowNet model for 2700 epochs with batch size 8. Following the Pytorch Implementation of FlowNet, we use the proposed data augmentation including translation, rotation, image crop, vertical flip, and horizontal flip. For the AutoFlow result, we use the published AutoFlow dataset (Sun et al., 2021). To generate the DFlow-GMA dataset, we replace the RAFT proxy networks with GMA networks (Jiang et al., 2021) while selecting the KITTI 2015 dataset as the target dataset. The set $\alpha$ to 13 for the DFlow-GMA dataset and pre-train the GMA network on DFlow-GMA with a data size of 2k.

---

[3]https://github.com/ClementPinard/FlowNetPytorch

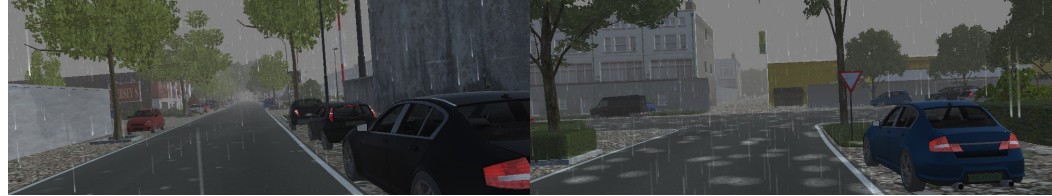

Figure 6: **Data samples of the rainy scene of Virtual KITTI.**

Table 9: **Contrasting analysis in pre-training.** The dataset generated with the contrasting effect achieves comparable results at the target dataset, KITTI 2015. The contrasting effect leads to a notable performance improvement on Sintel, which is not used for data generation.

| Contrasting effect | Evaluation dataset | | |
| --- | --- | --- | --- |
| | Sintel clean | Sintel final | KITTI 2015 |
| | AEPE | AEPE | AEPE / F1 |
| ✗ | 4.38 | 5.03 | 4.61 / **13.86** |
| ✓ | **2.14** | **3.79** | **4.31** / 14.29 |

*\*Bold denotes the best.*

### B.4 Details of Fine-tuning Results

To achieve fine-tuning results on the public benchmarks, we fine-tune the RAFT model pre-trained on DFlow. Except for the initial data schedule, *i.e.*, FlyingChairs followed by FlyingThings3D, we follow the same dataset schedule and training details of the original implementation (Teed & Deng, 2020).

### B.5 Details of Robustness against Corrupted Data

The photometric inconsistency is the main cause of the error in optical flow estimation. In real-world images, real-world effects, such as blur, fog, and illumination change, cause photometric inconsistency, and these effects frequently occur. Since DFlow generates a dataset using the compressed knowledge of the target dataset, we verify DFlow's ability to generate the dataset for training a robust optical flow network against photometric inconsistency. We use rainy scenes of Virtual KITTI (Gaidon et al., 2016) because the photometric inconsistency is dominant, as shown in Fig. 6. To obtain the target network, we further train the base network on the rainy scenes for 20k iterations and generate an additional dataset, DFlow-V, by targeting the rainy scenes. We use the published weight of RAFT trained on the RF-AB dataset for the result of RealFlow (Han et al., 2022).

## C Additional Experiments

We analyze the contrasting effect of base and target networks in Sec. C.1, and evaluate the validation performance with DFlow in Sec. C.2.

### C.1 Contrasting Effect of Base and Target Networks

To evaluate the contrasting effect of base and target networks, we measure the pre-training performance depending on the contrasting effect. The details will be presented in the following paragraphs.

**Contrasting Analysis in Pre-training.** For the contrasting analysis, we generate two types of dataset with a sample size of 2k. For the first dataset, we use KITTI 2015 (Menze & Geiger, 2015) as the target dataset, *i.e.*, the target network is obtained by fine-tuning the base network on the KITTI 2015 dataset. To synthesize the first dataset with contrasting effect, we minimize and maximizes the loss of the target and base networks, respectively. For the other dataset, we generate a new target network that is pre-trained on the KITTI 2015 dataset. With the new target network, we generate a new pre-training dataset that is synthesized to minimize the loss of the new target data network only, *i.e.*, the contrasting effect is not applied.

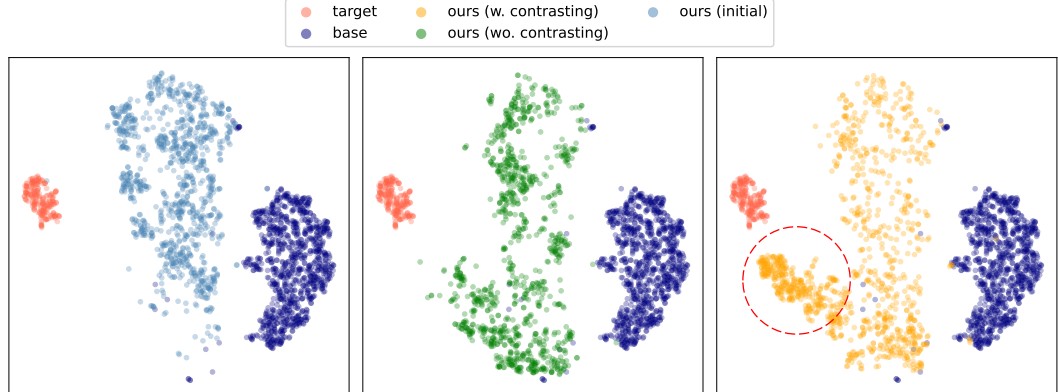

Figure 7: **t-SNE plot (**van der Maaten & Hinton, 2008**) of different datasets.** We use the context encoder of the base network trained on FlyingChairs as a feature extractor and extract the features of several datasets. (Red): target datasets, *e.g.*, KITTI 2015, (Blue): base dataset, *e.g.*, FlyingChairs, (Sky): our initial dataset before optimization, (green): updated our dataset without the contrasting effect, (Yellow): updated our dataset with the contrasting effect. The features of our dataset with the contrasting effect are getting closer to the features of the target dataset through the optimization process than the features of other datasets.

Table 9 shows the pre-training results depending on the contrasting effect. With the contrasting effect, the RAFT network achieves comparable results on the target dataset, *i.e.*, KITTI 2015, compared to the other network trained on the dataset generated with the new target network only. However, without the contrasting effect, the network shows notable performance degradation on Sintel (Butler et al., 2012), which is not used for data generation.

We also visualize data features to identify the effect of the contrasting effect. We embed the features of several datasets, such as FlyingChairs, KITTI 2015, our datasets with and without the contrasting effect, and our dataset before optimization. We use the context encoder of the base network as a feature extractor. As shown in Fig. 7, ours with the contrasting effect is closer to the target than one without the contrasting effect. We additionally com-

Table 10: **Distance of feature mean between ours and KITTI 2015.**

| Datasets | MMD |
|---|---|
| Ours wo. contrasting | 6.6892 |
| Ours w. contrasting | 6.1454 |

pute distances of feature mean between ours and KITTI 2015. As shown in Table 10, the maximum mean discrepancy (MMD) (Gretton et al., 2012) of ours with the contrasting effect is lower than ours without the contrasting effect. These results show that the contrasting effect drives the data generation toward an improving direction in a target domain.

## C.2 VALIDATION RESULTS

The advantage of the proposed pipeline is that the generated data can be used to improve the validation performance on the unseen target dataset. We split the KITTI 2015 dataset into 50 training samples and 150 validation samples; KITTI-50 and KITTI-150. Using the target network fine-tuned on KITTI-50, we synthesize a dataset; Ours$_{\text{KITTI-50}}$. Therefore, any data or information on KITTI-150 is not included in the data generation process. We fine-tune the base network on several combinations of datasets with 20k iterations and evaluate the validation performance on KITTI-150. As shown in Table 11, fine-tuning on the combination of Sintel and KITTI-50 shows comparable performance with the one of fine-tuning on KITTI-50 alone. In contrast, we observe notable improvement in validation performance when adding the same number of Ours$_{\text{KITTI-50}}$ to the training dataset. As our data generation pipeline is not limited to the number of samples that can generate, we analyze the effect of our data size and observe the improvement of validation accuracy.

Table 11: **Comparison of the validation performance of networks fine-tuned on different dataset combinations.** We fine-tune the base network on the combination of KITTI-50 and another dataset. Ours$_{\text{KITTI-50}}$ notably improves the validation accuracy on the unseen KITTI dataset, KITTI-150.

| | | Validation dataset |
|---|---|---|
| Fine-tune dataset | Data size | KITTI-150 |
| | | AEPE / F1 |
| KITTI-50 | 50 | 3.41 / 11.47 |
| KITTI-50+Sintel | 50+2082 | 3.39 / 11.21 |
| KITTI-50+Ours$_{\text{KITTI-50}}$ | 50+2082 | 3.07 / 10.99 |
| KITTI-50+Ours$_{\text{KITTI-50}}$ | 50+12003 | **2.98** / **10.70** |

*__Bold__ denotes the best, and underline denotes the second best.

Table 12: **Pre-training results depending on the amounts of DFlow data.** We train the RAFT networks on FlyingChairs (Fischer et al., 2015) and various amounts of DFlow dataset. We evaluate the performance on the Sintel and KITTI 2015 datasets.

| | | Evaluation dataset | | |
|---|---|---|---|---|
| Dataset | Data size | Sintel clean | Sintel final | KITTI 2015 |
| | | AEPE | AEPE | AEPE / F1 |
| FlyingChairs | 22873 | 2.28 | 4.51 | 9.85 / 37.56 |
| DFlow | 100 | 2.72 | 4.03 | 9.15 / 23.09 |
| DFlow | 500 | 2.20 | 3.48 | 5.40 / 17.54 |
| DFlow | 1000 | 2.02 | 3.24 | 5.22 / 17.06 |
| DFlow | 2000 | 1.90 | 3.02 | 4.90 / 15.78 |
| DFlow | 4000 | 1.87 | 2.92 | 4.78 / 15.64 |
| DFlow | 8000 | 1.81 | 2.91 | 4.88 / 15.51 |
| DFlow | 15000 | 1.81 | 2.93 | 4.59 / 15.03 |

## C.3    DATASET SIZE

The number of the dataset is a key factor for training accurate optical flow networks, and synthetic optical flow data have an advantage because of the annotation efficiency compared to the real-world data. We analyze the effect of data size in the optical flow network, RAFT (Teed & Deng, 2020) We train the RAFT network on the various amount of DFlow data and evaluate the performance on both Sintel and KITTI 2015 datasets. We determine FlyingChairs (Fischer et al., 2015) as a competing dataset because FlyingChairs and DFlow have the same data resolution. As shown in Table 12, the RAFT network trained on the 500 DFlow dataset outperforms the model trained on the full FlyingChairs dataset. This result shows that DFlow data has more information with the same data resolution. The overall performance on Sintel and KITTI 2015 datasets is also improved as the data size increase, which shows that diverse texture and motion is effective for training the optical flow network.

## C.4    PRE-TRAINING PERFORMANCE ACCORDING TO MOTION MAGNITUDES

Table 13 summarizes AEPEs of RAFT pre-trained on respective FlyingChairs (Fischer et al., 2015) and our DFlow according to different motion ranges. The model trained on DFlow shows higher accuracy than that of FlyingChairs except for the minimal motion range. Performance gaps between the models tend to be enlarged in larger motion. These results may be explained by human intuition that the small motion hardly contributes to the error of optical flow networks. For the same reason, most motion magnitudes of DFlow are in the middle and high motion ranges, as shown in Fig. 8.

Table 13: **Motion accuracy of the RAFT networks in different magnitude ranges.** We evaluate the AEPE score of models pre-trained on FlyingChairs and DFlow in different motion ranges. As the motion magnitude increases, the model trained on DFlow achieves more considerable performance improvements than that trained on FlyingChairs.

| Pre-training dataset | Evalutaion dataset | Motion magnitude ranges | | | | |
|---|---|---|---|---|---|---|
| | | $< 1$ | $[1, 10]$ | $(10, 20]$ | $(20, 30]$ | $> 30$ |
| FlyingChairs | Sintel Final | **0.57** | 1.15 | 3.31 | 6.47 | 24.08 |
| DFlow (Ours) | | 0.69 | **0.80** | **2.04** | **3.59** | **15.18** |
| FlyingChairs | KITTI 2015 | **0.74** | 1.67 | 2.65 | 3.86 | 21.32 |
| DFlow (Ours) | | 1.16 | **0.81** | **1.31** | **2.10** | **9.72** |

\*\***Bold** denotes the best.

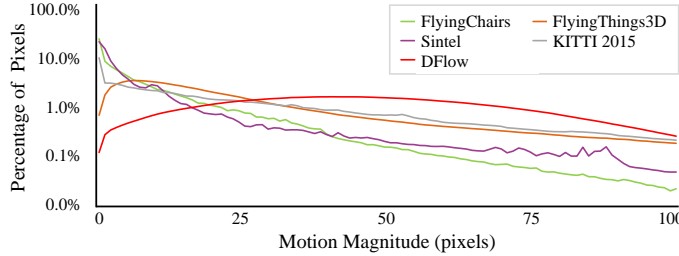

Figure 8: **Motion magnitude histograms.** DFlow focuses more on the mid-high ranges of motion than the other datasets, because small motion hardly contributes to errors of networks.

Table 14: **Analysis of multi-target datasets.** The target of DFlow is Sintel, DFlow-V rainy scenes of Virtual KITTI. We denote DFlow+DFlow-V as the multi-target datasets. Multi-target datasets show the trade-off in general but achieve the highest performance on KITTI 2015.

| Dataset | Evaluation dataset | | | |
|---|---|---|---|---|
| | Sintel clean | Sintel final | KITTI 2015 | Rainy scene |
| | AEPE | AEPE | AEPE / F1 | AEPE / F1 |
| DFlow | **1.81** | **2.93** | 4.59 / 15.03 | 5.87 / 22.89 |
| DFlow-V | 2.57 | 3.86 | 5.98 / 15.16 | **3.11** / **12.61** |
| DFlow + DFlow-V | 1.99 | 3.14 | **4.39** / **14.84** | 4.30 / 16.91 |

\***Bold** and underline note the best and second best, respectively.

## C.5 ANALYSIS OF MULTI-TARGET DATASETS

We focus on generating the dataset close to the target dataset. It is questionable whether multi-target datasets can be used to generate the DFlow dataset consisting of diverse data samples. The targets of DFlow and DFlow-V are Sintel and rainy scenes of Virtual KITTI. Table 14 lists the optical flow performance on Sintel clean, Sintel final, KITTI 2015, and the rainy scene of Virtual KITTI. As expected, DFlow-V achieves higher performance than DFlow on the rainy scene; but DFlow-V has inferior performance on the other datasets. When combining DFlow and DFlow-V, we observe that there is a trade-off; the performance of DFlow with DFlow-V is usually between the performance of DFlow and DFlow-V. However, the mixture of DFlow and DFlow-V outperforms in KITTI 2015. It indicates that exploiting multi-target datasets is a promising research direction.

C.6 ADDITIONAL QUALITATIVE RESULTS

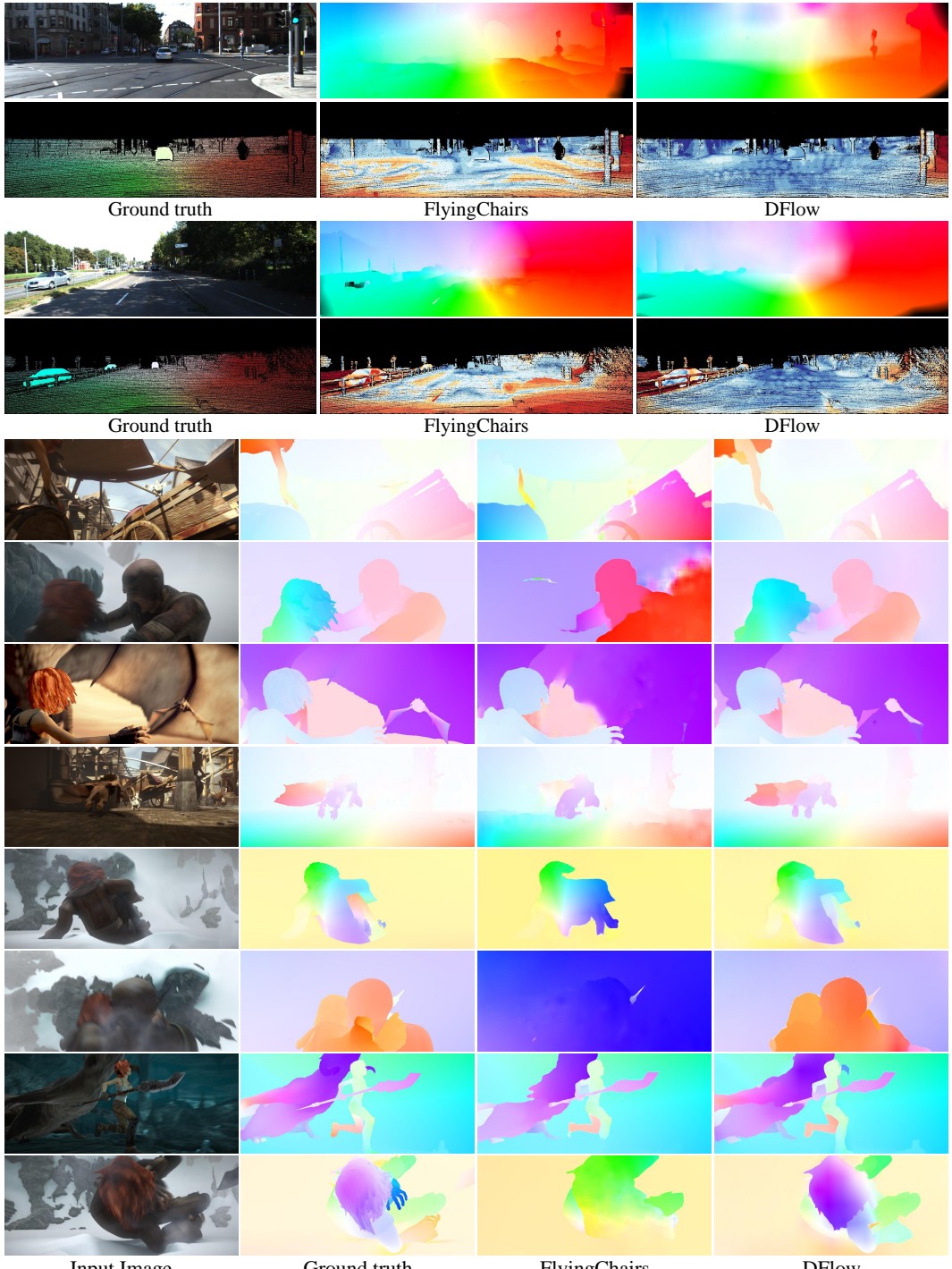

Figure 9: **Additional qualitative results.** Top: pre-training results on the KITTI 2015 dataset. Bottom: pre-training results on Sintel Final pass.

# D   ADDITIONAL GENERATED DATA SAMPLES

Figure 10 shows additional data samples. Color perturbation, texture noise, fog, and motion blur can be found in the data samples.

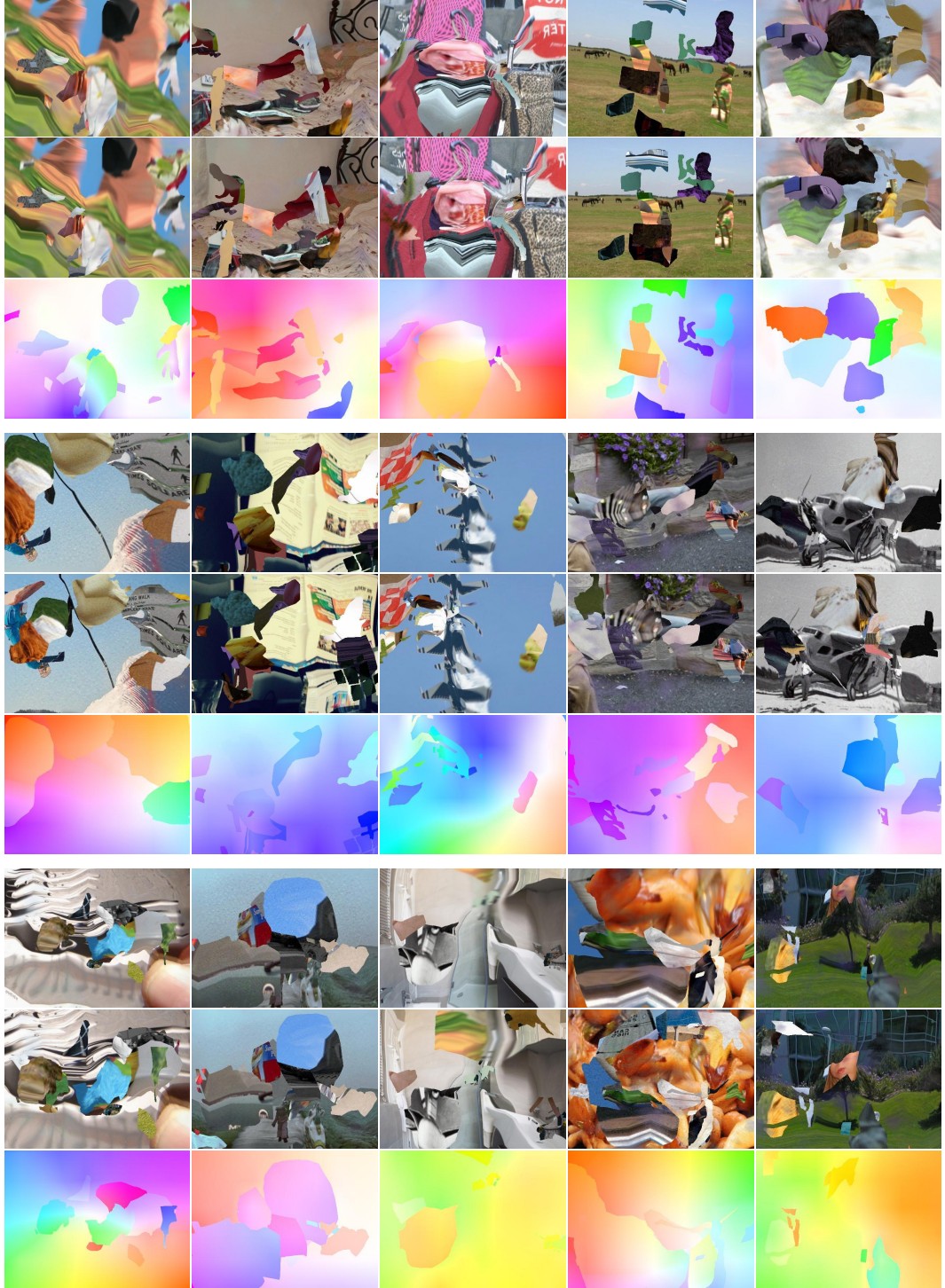

Figure 10: **Additional generated data samples.** Top: current frames; Middle: next frames; Bottom: visualizations of ground truth optical flow.

