# OpenReview forum: "DFlow: Learning to Synthesize Better Optical Flow Datasets via a Differentiable Pipeline"
_ICLR.cc/2023/Conference — ICLR 2023 poster_

### Official Review · Reviewer_dkLC · 2022-10-19

**Confidence:** 3
**Clarity, Quality, Novelty And Reproducibility:** 1. The presentation is not very clear…
**Correctness:** 3
**Technical Novelty And Significance:** 3
**Empirical Novelty And Significance:** 2
**Recommendation:** 5

**Strength And Weaknesses:**

Strength:
1. The fully differentiable data generation process is interesting and efficient which includes various components including color perturbation, geometric warping, flow field translation, layer composition and real-world effects.
2. The experiment on the analysis of each data generation component is clear and convincing.

Weakness:
1. The data generation process of DFlow needs a target dataset and a common base dataset on which the proxy network models are trained in fully supervised way, this makes the generated dataset somewhat task-dependent.
2. The declaration of SOTA performance on the Sintel public benchmark is not correct: recently there are lots of published optical flow algorithms achieved better performance than RAFT, please ref to: http://sintel.is.tue.mpg.de/results for more details. Table 5 shows comparison with RAFT but does not show SOTA performance, is it possible to apply DFlow to more recent deep network structures?
3. It is not clear whether the dataset size of DFlow affects the optical flow estimation result and whether the data generation process can use multiple target datasets for better generalization on pre-train or better fine-tune performance.

**Summary Of The Paper:**

This paper proposed a differentiable data generation pipeline for optical flow which depends on a target optical flow dataset, a common base dataset and prior data generation constraints on grid warping and texture noise. The data generation process is much more efficient than existing work of AutoFlow and the method achieves state-of-the-art performance on public Sintel benchmark.

**Summary Of The Review:**

The idea of differentiable data generation is interesting but the experiment result is not impressive and the reproducibility is not good.

---

> ### Author Response · Authors · 2022-11-19
> **Response to Reviewer dkLC's Review**
>
> We thank the reviewer for the detailed and constructive comments. We address the concerns and questions below.
>
> **Q1: The data generation process of DFlow needs a target dataset and a common base dataset on which the proxy network models are trained in fully supervised way, this makes the generated dataset somewhat task-dependent.**
>
> Note that the reviewers’ comment is also applicable to AutoFlow [B1] and other closely related works, where they focus on the task-dependent and specific case of optical flow. Thus, it would not harm the contribution of our work.
>
> To remind, it is our intention, motivation, and contribution. Our motivation is the real-world optical flow data that are typically hard to acquire, and recent models demand a high-quality and big dataset. Thus, our contribution specifically focuses on generating data for the optical flow task with extensive analyses.
>
> Of course, our framework may be generalized to other tasks, such as segmentation, stereo matching, etc., but which are out-of-scope for our work, in that dataset construction of these tasks is relatively easier than the optical flow data of natural scenes.
>
> However, it is a valuable comment, and we will consider an extension to other tasks which we think is a promising direction.
>
> [B1]: Sun, Deqing, et al., “Autoflow: Learning a better training set for optical flow”, Proceedings of the IEEE/CVF Conference on Computer Vision and Pattern Recognition (CVPR), 2021.
>
> **Q2: The declaration of SOTA performance on the Sintel public benchmark is not correct: recently there are lots of published optical flow algorithms achieved better performance than RAFT, please ref to: [http://sintel.is.tue.mpg.de/results](http://sintel.is.tue.mpg.de/results) for more details. Table 5 shows comparison with RAFT but does not show SOTA performance.**
>
> First, we have revised to remove “state-of-the-art” and tone down the sentence and similar statements across all the parts in this revision to avoid related confusion (See abstract and page 7). Our original intention was that our method better performs among other dataset papers (including the AutoFlow dataset) under fair experiment settings, rather than architecture works. We have added an additional elaboration to clarify the evaluation setup (See Table 4 on page 8).
>
> Following the same setup suggested by our closest work, AutoFlow [B1], we use RAFT [B2] as the standard network, and compare our work with other dataset papers.
>
> [B2] Teed, Zachary, and Jia Deng., “Raft: Recurrent all-pairs field transforms for optical flow”, European conference on computer vision (ECCV), 2020.
>
> **Q3: Is it possible to apply DFlow to more recent deep network structures?**
>
> It would be interesting to generate other DFlow datasets by using the recent optical flow networks. We think that our data generator is agnostic to the deep network structure, but leave the experiment for a journal extension due to the limited period of rebuttal.
>
> **Q4: It is not clear whether the dataset size of DFlow affects the optical flow estimation result.**
>
> Thanks for the constructive comment. We have newly added the suggested experiment in Sec. C.3 (page 19) in the appendix. The results of the experiment are listed below.
>
> | Data size | Sintel Clean | Sintel Final | KITTI 2015 |
> | --- | --- | --- | --- |
> | 100 | 2.72 | 4.03 | 9.15 / 23.09 |
> | 500 | 2.20 | 3.48 | 5.40 / 17.54 |
> | 1000 | 2.02 | 3.24 | 5.22 / 17.06 |
> | 2000 | 1.90 | 3.02 | 4.90 / 15.78 |
> | 4000 | 1.87 | 2.92 | 4.78 / 15.64 |
> | 8000 | 1.81 | 2.91 | 4.88 / 15.51 |
> | 15000 | 1.81 | 2.93 | 4.59 / 15.03 |
>
> We observe that the larger the dataset of DFlow is, the more improvement is.

---

> > ### Author Response · Authors · 2022-11-19
> > **Response to Reviewer dkLC's Review**
> >
> > **Q5: whether the data generation process can use multiple target datasets for better generalization on pre-train or better fine-tune performance.**
> >
> > We have added the suggested experiment in Sec. C.5 (page 21) in the appendix.
> >
> > The suggestion of the reviewer is interesting, and implies asking about the existence of a meta-dataset for better generalization and better fine-tuning performance across multiple target datasets.
> >
> > Given limited resources of time and computation, we have conducted a manageable-size experiment that may hint at the above question. We select the Sintel and rainy scenes of Virtual KITTI for the distinct characteristics of target datasets (See Figure 7 on page 17). Sintel is based on the 3D animated movie. The rainy scenes of Virtual KITTI simulate the driving scenes while the rain effect is dominant, which violates the photometric consistency.
> >
> > We generate two DFlow datasets using Sintel and rainy scenes of Virtual KITTI. We denote them as DFlow-S and DFlow-V, respectively. Combining with two datasets, we train the RAFT network and list the performance on KITTI 2015 below.
> >
> > |  | Sintel Clean | Sintel Final | KITTI 2015 | Rainy Scene |
> > | --- | --- | --- | --- | --- |
> > | DFlow-S | 1.81 | 2.93 | 4.59 / 15.03 | 5.87 / 22.89 |
> > | DFlow-V | 2.57 | 3.86 | 5.98 / 15.16 | 3.11 / 12.61 |
> > | DFlow-V + DFlow-S | 1.99 | 3.14 | 4.39 / 14.84 | 4.30 / 16.91 |
> >
> > Interestingly, the performance on the unseen KITTI 2015 dataset with both datasets is more improved, compared to the ones with the respective dataset. We think that information of multiple target datasets could be complementary with each other, and may lead to the performance improvement.
> >
> > **Q6: The presentation is not very clear and sometimes confusing, i.e. in Sec 3.1 page 4 "We hypothesize that, since a snippet of the target dataset is typically on small-scale in general scenarios", and "For example, KITTI 2015 consists of just 200 number of training samples", but the DFlow's target dataset is Sintel which is not small.**
> >
> > First, we have clarified the part in Sec. 3.1 (page 4) as follows:
> >
> > "We assume that target networks trained on the popular benchmarks (Sintel, KITTI 2015) may not produce sufficiently rich training signals, because those benckmarks are notably smaller than synthetic datasets (FlyingChairs, FlyingThings3D) widely used in pre-training.”
> >
> > In addition, the reviewer mentioned that Sintel is not a small dataset. We would like to correct the misunderstanding.
> >
> > Our method is motivated by the annotation difficulty of real-world optical flow data, and the difficulty typically occurs in the real-world; thus, we intend the motivation sentences in Sec. 3.1 page 4 to be applied to real-world datasets, NOT synthetic datasets like Sintel.
> >
> > Furthermore, compared to the FlyingChairs and FlyingThings3D, which are synthetic datasets widely used in pre-training, Sintel is almost 5% size of FlyingThings3D (Refer to the following table for dataset size comparison)
> >
> > | Dataset | Train Data Size |
> > | --- | --- |
> > | FlyingThings3D (2017, synthetic) | 40302 |
> > | FlyingChairs (2015, synthetic) | 22872 |
> > | Virtual KITTI (2016, synthetic) | 21260 |
> > | VIPER (2017, synthetic) | 13359 |
> > | Sintel (2012, synthetic) | 2082 |
> > | HD1K (2016, real) | 1047 |
> > | KITTI 2015 (2015, real) | 200 |
> >
> > Our closest work, AutoFlow, also evaluate their dataset on Sintel benchmark. To compare with AutoFlow, we also validate our approach on Sintel as the target dataset.
> >
> > **Q7: In Sec. 3.3. "Implementation details can be found in the appendix", it would be good to add "A.5"**
> >
> > We appreciate again the review to make our paper clearer. We have revised the sentence and have applied that to make the paper consistent**.**
> >
> > **Q8: In Table 3. the DFlow (Ours) targeted on Sintel achieved best performance on KITTI 2015, but worse performance on Sintel final, it would be good to add a few sentences for explanation.**
> >
> > We have newly added the discussion as the reviewer’s suggestion in Sec. 4 (page 6).
> >
> > We use real-world images as background and foreground texture sources for data sample generation in DFlow. We think that the real-world textures affect the performance on KITTI2015.
> >
> > Also, we train the target model with both Sintel clean and final when generating a dataset in DFlow. This is difference from AutoFlow which used Sintel final alone. Thus, as a trade-off of using both splits of Sintel clean and final, the performance on Sintel final might be a bit degraded.
> >
> > **Q9: The novelty of the proposed method is OK but the reproducibility is not good since the training of data generation pipeline is complex and may need considerable parameter tuning and training adjustment, public open of training source code is necessary for good reproducibility.**
> >
> > We thank the reviewer for acknowledging the novelty of our method. For reproducibility, we will release our data generation and training code to the public if accepted.

---

> > > ### Author Response · Authors · 2022-12-08
> > > **Response to Reviewer dkLC's Review**
> > >
> > > **Q3: Is it possible to apply DFlow to more recent deep network structures?**
> > >
> > > According to the reviewer's question, we have newly conducted a generalization test of our dataset approach with the GMA network [B3], which is a more recent variant of the RAFT network. In this experiment, we select the KITTI 2015 dataset as the target dataset and generate the DFlow-GMA dataset using the GMA network. Using our DFlow-GMA dataset, we train the GMA network and list the performance on KITTI 2015 below.
> > >
> > > | Networks | Dataset | Data size | KITTI 2015 (AEPE / F1) |
> > > | --- | --- | --- | --- |
> > > | GMA [B3] | FlyingChairs→FlyingThings3D | 22k→40k | 4.69 / 17.10 |
> > > | GMA [B3] | DFlow-GMA | 2k | 4.47 / 13.10 |
> > >
> > > The GMA network trained on our DFlow-GMA achieves higher performance than the one trained on FlyingChairs followed by FlyingThings3D. From this observation, we postulate that our method is agnostic to the deep network structure and can be applied to the recent network than RAFT. Note that we also showed the generality of our DFlow dataset approach by showing that the dataset generated by RAFT (DFlow-RAFT) is also effective in training FlowNet in Table 3 at page 7 (i.e., transferability). With the results in Table 3 and the above new experiment, we hope the reviewer finds the fact that our approach is generalized to other architectures. We will add the additional results and discussion in the final version if accepted. Thanks.
> > >
> > > [B3] Jiang, Shihao, et al. "Learning to estimate hidden motions with global motion aggregation." Proceedings of the IEEE/CVF International Conference on Computer Vision (ICCV). 2021.

---

### Official Review · Reviewer_tPfm · 2022-10-25

**Confidence:** 4
**Correctness:** 3
**Technical Novelty And Significance:** 3
**Empirical Novelty And Significance:** 3
**Recommendation:** 8

**Clarity, Quality, Novelty And Reproducibility:**

__Clarity__

- (In Section 3) Optimization: I wonder how optimization works. Does the method jointly optimize network parameters and dataset parameters together? I didn't fully understand how they work. For example on Page 4, the paper introduces a base network. How the two networks and the parameters are optimized together? It would be good if any pseudo source code is provided for an explanation (for example, at each step, what is optimized, which training dataset is used, which test dataset is used for evaluation, etc..)

- (Eq. 6) Is there any ablation study for the layer decomposition? (Softmax splatting vs. alpha blending)

- (Just an idea) (Table 6) Is it possible to optimize the binary setups in a differentiable manner? (e.g., learning a continuous variable between 0 and 1 as a probability for the On or Off selection).  It's because some binary setups show different results on different datasets (e.g., Noise regularization, Fog, Motion blur, etc..), it would be also great if those are optimizable as well.

---
__Others__

- Quality: Good quality. Sufficient figures and numbers for a clear understanding of the paper


- Novelty: Somewhat significant, considering its performance in general.

- Reproducibility: Despite clear details included, it would really help to have source code to reproduce the result. It may be difficult without it.

**Strength And Weaknesses:**

__Strength__

- Computational efficiency
  Comparing with AutoFlow, the method is substantially more efficient (ref. Table 1). It enables the algorithm to run on a small-scale experiment setup.

- Detail ablation study
  The paper provides clear & detail ablation study for each design choice. It makes paper more transparent.

---

__Weakness__

- I don't see any significant weakness of the paper so far. Rather there are some details that are better to be clarified. (continued in the following section).

**Summary Of The Paper:**

The paper introduces a differentiable pipeline for learning an optical flow dataset. Given a pre-defined set of dataset generation parameters and network, the method optimizes the variable using an optical flow loss in an end-to-end manner. Comparing to the previous work, the method achieves competitive accuracy but mostly better accuracy. Especially in terms of computational cost (e.g., training time), the method is an order of magnitude more efficient than the previous work.

**Summary Of The Review:**

The paper shows empirically good results, great efficiency, and it provides detailed ablation study. However, it was difficult for me to understand some technical parts (see the clarity section above). I would like to raise my rating after clearing up those concerns through a discussion phase.

----

__(Post discussion)__

I would like to raise my rating to '8: accept, good paper'. The authors' response resolved my main concern about the clarity of the paper. I also read other reviews and responses on them, and they seem fairly well addressed. The paper mainly demonstrates empirically good accuracy over the direct competitor (AutoFlow) with better training efficiency.

---

> ### Author Response · Authors · 2022-11-19
> **Response to Reviewer tPfm's Review**
>
> Thanks for the reviewer for acknowledging the computational efficiency of our method and detailed ablation study. We address the concerns below
>
> **Q1: Optimization: I wonder how optimization works. Does the method jointly optimize network parameters and dataset parameters together? I didn't fully understand how they work. For example on Page 4, the paper introduces a base network. How the two networks and the parameters are optimized together? It would be good if any pseudo source code is provided for an explanation (for example, at each step, what are optimized, which training dataset is used, which test dataset is used for evaluation, etc..)**
>
> We have added the summarization of our method in Sec. 3 (page. 6) and pseudo-code in Sec.3 (page 6) and Sec. A.2 (page. 15). Also, while we missed in the initial submission, we will publish our code and dataset for reproducibility if accepted.
>
> We provide the summarization of our method below.
>
> - Dataset setting
>     - Base dataset: FlyingChairs
>     - Target dataset: Sintel
>     - Evaluation dataset: Sintel, KITTI2015
>
> [Procedure]
>
> 1. We prepare two networks trained on base and target datasets, respectively. We assume that the target network embeds the information of the target dataset.
> 2. With the fixed two trained networks, we update the parameters of the data generator in a contrastive way while not updating the weight parameters of the two networks. It is done by minimizing the loss in terms of the target network and maximizing the loss in terms of the base network. This is the generation step to synthesize a new synthetic dataset aiming at the target dataset.
> 3. Using the generated dataset, we train the optical flow network.
>
> Note that we do not optimize the parameters of networks and datasets together.
>
> **Q2: (Eq. 6) Is there any ablation study for the layer decomposition? (Softmax splatting vs. alpha blending)**
>
> We thank the reviewer for suggesting the insightful ablation study. To reflect the suggestion, we have newly added the suggested experiment (See Table 5 on page 8). The result is listed below.
>
> | Method | Sintel Clean / Final | KITTI 2015 |
> | --- | --- | --- |
> | Softmax splatting | 1.86 / 3.04 | 5.21 & 16.27 |
> | Alpha blending | 1.95 / 2.83 | 5.23 & 16.06 |
>
> We observe the following trade-off.
>
> - The softmax splatting improves the performance in Sintel clean more than the alpha blending.
> - The alpha blending performs better in Sintel final than the softmax splitting.
>
> We think that it would improve the quality of our paper.
>
> **Q3: (Just an idea) (Table 6) Is it possible to optimize the binary setups in a differentiable manner? (e.g., learning a continuous variable between 0 and 1 as a probability for the On or Off selection). It's because some binary setups show different results on different datasets (e.g., Noise regularization, Fog, Motion blur, etc..), it would be also great if those are optimizable as well.**
>
> This is another interesting idea. Our method does not have a switch that turns on or off the real-world effects, but the parameterized data generation modules have abilities to adjust the strength of each effect to some extent.
>
> Nonetheless, explicitly introducing switching variables can be modeled by differentiable softmax. Furthermore, we may introduce stochastic variables, e.g., using Gumbel-softmax. Regardless of actual effectiveness, those would be promising directions to be explored, because the real-world effects are complex and contain uncertainty.
>
> We thank you again for the constructive comment and leave the experiment for future research due to the limited period of rebuttal.
>
> **Q4: However, it was difficult for me to understand some technical parts (see clarity section above). I would like to raise my rating after clearing up those concerns through a discussion phase.**
>
> As discussed in Q2, we have revised our paper containing a clear summarization of our pipeline and the pseudo source code in the appendix to help the reader better understand our work.
>
> We appreciate the valuable comments to improve our paper.

---

> > ### Comment · Reviewer_tPfm · 2022-11-25
> > **Response**
> >
> > Thank you very much for clarifying the details! Now I have a clear understanding of the paper. But before raising the rating, I would like to double-check my understanding. Could you maybe confirm that the following summarization is correct?
> >
> > - The target network is a differentiable proxy of the target dataset: compressing (or distilling) information of the target dataset into the network.
> > - Then, the compressed (or distilled) information from the frozen target network will be propagated into the dataset generator by optimizing the data generation parameters.
> > - Using the contrastive loss with the base network helps stabilize the training.
> >
> >
> > Thank you,
> > Reviewer tPfm

---

> > > ### Author Response · Authors · 2022-11-27
> > > **Response to Reviewer tPfm's Response**
> > >
> > > We thank the reviewer for the response.
> > > We read the suggested summarization and the summarization is absolutely correct.
> > > We appreciate the reviewer’s valuable comment again to clarify our method in our paper.

---

> > > > ### Comment · Reviewer_tPfm · 2022-11-30
> > > > **Response**
> > > >
> > > > Thank you for the clarification! By the way, while reading the paper, there are still some concerns that I couldn't find answers from the paper.
> > > >
> > > > - How to avoid shortcuts (or trivial solution)?: The loss is evaluated on the rendered dataset, so one of the obvious shortcuts would be to generate very easy examples for the network (e.g., no heavy blur/fog augmentation, and nearly zero motion from geometric warping or flow translation). I wonder how the method avoids such shortcuts during the optimization? It seems that the method is still able to avoid such solutions without the contrastive loss (in Table 9).
> > > > - How's the learning rate when optimizing $\theta$?
> > > > - Initialization: From the attached video (visualization of optimization) in the supplementary, the initial example looks already challenging enough to be a good example for evaluation. How are the hyper-parameters ($\theta$) are initialized? or how does the method initialize the examples prior to the optimization?
> > > > - Detail parameterization: It was difficult to find detail parameterizations of all augmentation schemes (grid-warping, rigid transformation, noise, fog, blur, etc.). It would be great to provide such details (such as equations) for a better reproducibility. Does each data sample have its own set of $\theta$s and they are optimized per example? or are the $\theta$s kinds of control parameters for random distributions per target dataset, and globally optimized for the target dataset?

---

> > > > > ### Author Response · Authors · 2022-12-05
> > > > > **Response to Reviewer tPfm's Response**
> > > > >
> > > > > Dear Reviewer tPfm, thanks for first raising the score to “8: Accept”, but it is also unfortunate that the reviewer decreased the score to “6: marginally above the acceptance threshold“ with new additional questions mainly related to the reproducibility and parameters. First of all, as aforementioned in the previous response, ***we will publish our code and dataset for reproducibility if accepted***.
> > > > >
> > > > > **Q1: How to avoid shortcuts (or trivial solution)?: The loss is evaluated on the rendered dataset, so one of the obvious shortcuts would be to generate very easy examples for the network (e.g., no heavy blur/fog augmentation, and nearly zero motion from geometric warping or flow translation). I wonder how the method avoids such shortcuts during the optimization? It seems that the method is still able to avoid such solutions without the contrastive loss (in Table 9).**
> > > > >
> > > > > We have not empirically observed such trivial solution cases, e.g., nearly zero motion, etc. Following the implementation of [AutoFlow], we sample parameter θ for each data from the uniform distribution with pre-specified ranges. We postulate that this random parameter initialization from the uniform distribution has a major effect on avoiding converging to the trivial solution, as shown in Table 9 (page 18).
> > > > >
> > > > > Also, since our method is based on gradient-based optimization, our method starts from an initial data parameter θ and searches for a better solution around the initial data point, i.e., a local minimum. In our observation, the resulting data converged to lower loss values than their corresponding initial data. Just in case the resulting data return higher loss values than 25 of the target network’s loss value, we discard those data (Section B.1 page 17).
> > > > >
> > > > > **Q2: How's the learning rate when optimizing θ?**
> > > > >
> > > > > We set the learning rate of real-world effects as 0.03 and the others as {1.0, 0.1, 0.01} depending on whether they are pixel-unit operations (e.g., translation parameters) or not. We decay the learning rates linearly over the update iterations; the decay factor is (1 - iter/80), where 80 is the maximum iteration of updates (Section B.1 page 17). We distinguish the pixel-unit operations and the others as:
> > > > >
> > > > > - Pixel-unit operation (learning rate of 1.0): translation of the rigid transformation, grid warping,  perspective warping, and translation of the flow field translation
> > > > > - Non-pixel-unit operation (learning rates of {0.1, 0.01}): rotation and scaling of the rigid transformation
> > > > >
> > > > >
> > > > >
> > > > > **Q3: Initialization: From the attached video (visualization of optimization) in the supplementary, the initial example looks already challenging enough to be a good example for evaluation. How are the hyper-parameters (θ) are initialized? or how does the method initialize the examples prior to the optimization?**
> > > > >
> > > > > As aforementioned in question Q1 of the second question list, following the implementation of [AutoFlow], we sample the data parameters θ from the uniform distributions.
> > > > >
> > > > > While random initial data visually appear good enough, we observed that the RAFT network trained on the initial data shows poor accuracy on both Sintel and KITTI 2015, as shown in the data update analysis in Table 5 (page 8).
> > > > >
> > > > > For completeness, we list the initialization of each component below, which will be included in the appendix if accepted:
> > > > >
> > > > > ***Details of the initialization***
> > > > >
> > > > > - Color perturbation → White balance → [0, 1]
> > > > > - Geometric warping → Rigid transformation (translation, rotation, scaling) → ([-80, 80], [-5, 5], [0.75, 1.13])
> > > > > - Geometric warping → Grid warping → 0
> > > > > - Geometric warping → Perspective warping → [-25, 25]
> > > > > - Flow translation → Translation → [-50, 50]
> > > > > - Real-world effects → Texture noise → [-0.01, 0.01]
> > > > > - Real-world effects → Fog (fog color) → [0, 1]
> > > > > - Real-world effects → Motion blur (std of axis x, std of axis y, angle of rotation)  → ([1, 2], [3, 11], [0, 90])
> > > > >
> > > > > where [-a, b] denote the random number uniformly sampled from -a to b.

---

> > > > > > ### Author Response · Authors · 2022-12-05
> > > > > > **Response to Reviewer tPfm's Response**
> > > > > >
> > > > > >
> > > > > > **Q4: Detail parameterization: It was difficult to find detail parameterizations of all augmentation schemes (grid-warping, rigid transformation, noise, fog, blur, etc.). It would be great to provide such details for a better reproducibility. Does each data sample have its own set of θs and they are optimized per example? or are the θs kinds of control parameters for random distributions per target dataset, and globally optimized for the target dataset?**
> > > > > >
> > > > > > For the details of parameterization, please refer to the already presented information in Figure 2 (page 5) and Section A.3-A.5 in the appendix about the specific dimension of learnable parameters and details of data parameter **θ**, which would give some of the answers about the questions.
> > > > > >
> > > > > > However, by the reviewer’s question, we find that someone may feel that it would not be enough for reproducibility. By taking this chance, we list up more details, and the following details will be added to the camera-ready if accepted. Again, as aforementioned, we will release the code and generated dataset, so that no reproducibility concern is raised.
> > > > > >
> > > > > > We optimize the data parameter θ per example, not per target dataset.
> > > > > >
> > > > > > We follow the details of [AutoFlow] to parameterize the geometric warping, including the rigid transformation, grid-warping, and perspective warping. The rigid transformation (5 degrees of freedom; d.o.f) consists of translation (2 d.o.f), rotation (1 d.o.f), and scaling (2 d.o.f). Grid warping is the vertices of the 8x12 grid, which is interpolated to be added to the warping field. The perspective warping is simply parameterized by moving vertices of the 2x2 grid (8 d.o.f), i.e., the corners of the image.
> > > > > >
> > > > > > To simulate the motion blur and fog effects, we follow the implementation details of [AutoFlow]. We alpha-blend the foggy and motion-blurred frames to the frames after the layer composition. The learnable parameters of fog and motion blur are listed in Section A.5 (page 16)
> > > > > >
> > > > > > The texture noises are additive noises per frame. As described in Section A.5 (page 16), the random texture noise is first applied per pixel independently and optimized the per-pixel additive noise values directly. We apply two independent noises to the subsequent frames.
> > > > > >
> > > > > > Color perturbation consists of N white balance transformation for the N layer images, which is parameterized by three-channel scaling values (Section A.5, page 15). Each white balance is multiplied to each layer image.
> > > > > >
> > > > > > Flow field translation is a global translation parameter that translates the anchor geometric warping (Section 3.2, page 5).
> > > > > >
> > > > > > We thank the reviewer again for suggesting the details to be clarified. We will add the above description and more details in the camera-ready with our code if accepted.

---

> > > > > > > ### Comment · Reviewer_tPfm · 2022-12-08
> > > > > > > **Response**
> > > > > > >
> > > > > > > Thank you for your detailed clarification. Now I am able to understand the paper much better! My misunderstanding was that hyper-parameters for random sampling were optimized; actually, each data sample is optimized.
> > > > > > > (If the paper is accepted) It would be appreciated if the final version briefly includes the details in the response above or makes the texts clearer.

---

> > > > > > > > ### Author Response · Authors · 2022-12-08
> > > > > > > > **Thank you for championing our work**
> > > > > > > >
> > > > > > > > Thank the reviewer for raising the score. It is a pleasure to clarify the reviewer's questions. As the reviewer’s suggestion, we will reflect the details in the final version.

---

### Official Review · Reviewer_iPLs · 2022-10-25

**Confidence:** 3
**Correctness:** 3
**Technical Novelty And Significance:** 2
**Empirical Novelty And Significance:** Not applicable
**Recommendation:** 6

**Clarity, Quality, Novelty And Reproducibility:**

The overall quality is good.
Paper Clarity is good.
Technical novelty is marginal but makes sense for the target problem.
It can be reproduced.

**Strength And Weaknesses:**

Strength: the problem this work focuses on is meaningful and useful. The authors do propose a solid and effective solution for optical flow data generation.

Weakness: Limiation and future work should be discussed.

**Summary Of The Paper:**

This paper proposes a new data generation pipeline for training optical flow networks. The pipeline consists of parameterized differentiably geometric warping, flow field translation, color perturbation, and real-world effects. It also proposes a new objective function that drives the data optimization by leveraging the compressed knowledge of the proxy networks pre-trained on target and base datasets, respectively. Optical flow models trained on the datasets achieve favorable or superior performance against the competing datasets on pre-training and fine-tuning experiments.

**Summary Of The Review:**

My overall rating for this work is positive. The solid and comprehensive experimental results show the method and the generated datasets would benefits the community.

---

> ### Author Response · Authors · 2022-11-19
> **Response to Reviewer iPLs's Review**
>
> We thank the reviewer for the time and for acknowledging our work. We address the concern below
>
> **Q1: Limiation and future work should be discussed**
>
> Thanks for the comment. We have added the limitation and future work in Sec.5 (page. 9).
>
> We discuss the limitation and future work as follows.
>
> First, we use pre-defined data parametric generation operations, e.g., fog, isometric blur, geometric warping, etc. While our experiments show that our generated data are effective in the real-world dataset (KITTI 2015), the pre-defined and restricted operations might not span all the real-world effects. Thus, adding more diverse and complementary operations would further improve expressiveness and may lead to additional performance improvement. It is a promising way and would be worthwhile to investigate in the future.
>
> Second, our focus of this work is synthetic data generation aiming at a target dataset, and we approximate the target dataset with a proxy target model. For training the target model, we use optical flow supervision in the target dataset; thus, at least some amount of supervised data is required in our current method. We think that it would be an interesting research direction to investigate unsupervised learning to train the target model that is effective for our data generation pipeline.

---

### Official Review · Reviewer_uY3h · 2022-10-26

**Confidence:** 5
**Correctness:** 3
**Technical Novelty And Significance:** 3
**Empirical Novelty And Significance:** 3
**Recommendation:** 5

**Clarity, Quality, Novelty And Reproducibility:**

Clarity: The paper writing is general clear.

Quality: The paper quality is good in general.

Novelty: The paper owns its novelty in generating optical flow dataset. However the benefits of the new dataset has not been fully validated.

Reproducibility: The proposed method is relatively complex. It is not very easy to reproduce the results.


**Strength And Weaknesses:**

Strength:

+ A simple and efficient differentiable data generation pipeline for optical flow.

+ A contrastive-style learning scheme and its loss function by approximating expensive dataset-todataset comparison to leverage proxy neural networks.

Weaknesses:

- The claim that "the RAFT model pre-trained with DFlow achieves state-of-the-art performance on the Sintel public benchmark in fine-tuning" is questionable. According to the benchmark in Sintel. The performance of "DF-RAFT" is even not as good as the orignal RAFT.

- The paper did not discuss the limitations of the proposed approach.

- There are some recent work generating optical flow dataset in a rather cheap way such as :
RealFlow: EM-Based Realistic Optical Flow Dataset Generation from Videos, ECCV 2022.
Yunhui Han, Kunming Luo, Ao Luo, Jiangyu Liu, Haoqiang Fan, Guiming Luo, Shuaicheng Liu

**Summary Of The Paper:**

The paper proposed a differentiable optical flow data generation pipeline and a loss function to drive the pipeline. The proposed modules enable automatic and efficient synthesis of a dataset effectively to a target domain, given a snippet of target data. This distinctiveness is achieved by proposing an efficient data comparison method. Experiments show the competitive performance of DFlow against the prior arts in pre-training.

**Summary Of The Review:**

The paper proposed a differentiable optical flow data generation pipeline (DFlow) and a loss function to drive the pipeline. This is a simple and efficient differentiable data generation pipeline for optical flow. It also proposed a contrastive-style learning scheme and its loss function by approximating expensive dataset-todataset comparison to leverage proxy neural networks.

---

> ### Author Response · Authors · 2022-11-19
> **Response to Reviewer uY3h's Review**
>
> We thank the reviewer uY3h for the time and constructive comments. We discuss the concerns and questions of the reviewer below.
>
> **Q1: The claim that "the RAFT model pre-trained with DFlow achieves state-of-the-art performance on the Sintel public benchmark in fine-tuning" is questionable. According to the benchmark in Sintel. The performance of "DF-RAFT" is even not as good as the orignal RAFT.**
>
> First, we have revised to remove “state-of-the-art” and tone down the sentence and similar statements across all the parts in this revision to avoid related confusion (See abstract and page 7). Our original intention was to our method among the closely related dataset papers (including the AutoFlow dataset) under the fair experiment settings in fine-tuning. We have added an additional elaboration to clarify the evaluation setup (See Table 4 on page 8).
>
> For the performance of the original RAFT, we’d like to clarify the misleading point raised by the reviewer.
>
> The Sintel benchmark results of the original RAFT on the leaderboard were evaluated by using the warm-start initialization technique (utilizing the previous flow prediction as the next flow prediction initialization), which we and AutoFlow [A2] do not use. According to the same experimental protocol by our closest work [A2], we report the performance without the warm-start initialization method for the Sintel benchmark.
>
> The below table is the reported performance in the original RAFT [A1], which shows the effect of the warm-start technique.
>
> **Table 1 in RAFT [A1].** Sintel Clean and Final performance of RAFT with 2-view and warm-start.
>
> | Method | Sintel Clean | Sintel Final |
> | --- | --- | --- |
> | RAFT w/o warm-start | 1.94 | 3.18 |
> | RAFT with warm-start | 1.61 | 2.86 |
>
> [A1] Teed, Zachary, and Jia Deng., “Raft: Recurrent all-pairs field transforms for optical flow”, European conference on computer vision (ECCV), 2020.
>
> [A2]: Sun, Deqing, et al., “Autoflow: Learning a better training set for optical flow”, Proceedings of the IEEE/CVF Conference on Computer Vision and Pattern Recognition (CVPR), 2021.
>
> **Q2: The paper did not discuss the limitations of the proposed approach.**
>
> We have revised the paper to include the discussion about limitations in Sec. 5 (page. 9).
>
> We discuss the following limitations.
>
> 1.  We use the pre-defined elementary data generation operations, e.g., fog, geometric warping, etc. While our experiments show that our generated data are effective in the real-world dataset (KITTI 2015), the pre-defined and restricted operations might not span all the real-world effects. Thus, adding more diverse and complementary operations would further improve expressiveness and may lead to additional performance improvement.
> 2. Our method aims at a specific target dataset. For that, we need a target network trained on the target dataset, which requires at least some amount of optical flow annotations. To mitigate the requirement of the supervised data, it would be an interesting direction to investigate the way to train the target network in an unsupervised method, such that the unsupervisedly trained target network is effective for our data generation pipeline.
>
> **Q3: There are some recent work generating optical flow dataset in a rather cheap way such as : RealFlow: EM-Based Realistic Optical Flow Dataset Generation from Videos, ECCV 2022. Yunhui Han, Kunming Luo, Ao Luo, Jiangyu Liu, Haoqiang Fan, Guiming Luo, Shuaicheng Liu.**
>
> We thank the reviewer for suggesting the relevant work. We have cited RealFlow in Sec.2 (page. 2) and discussed the efficiency of RealFlow in the main paper in Sec.2 (page. 2).
>
> Separately, we would like to ask the reviewer to find that RealFlow is a concurrent work with ours, because it is published in ECCV (according to the publication date of ECCV) after the deadline of this ICLR submission.
>
> **Q4: The proposed method is relatively complex. It is not very easy to reproduce the results.**
>
> For reproducibility, we will publish the dataset and code used for data generation if accepted. We missed the promise about this in the initial submission. We apologize.

---

> > ### Author Response · Authors · 2022-11-30
> > **Response to Reviewer uY3h's Review**
> >
> > As aforementioned, while RealFlow is a concurrent work with this submission, we have conducted an additional experiment about RealFlow with our DFlow.
> >
> > We hypothesized that they would have complementary aspects due to their different design principles. RealFlow relies on iterative pseudo labeling of real data, while our DFlow learns parameters for generating synthetic data by constrasting base and target domains.
> >
> > Considering these characteristics, we have conducted a new experiment to combine the RealFlow and DFlow (ours) approaches as follows:
> >
> > Dataset setting
> >
> > - Base dataset: RF-Ktrain [RealFlow]
> > - Target dataset: KITTI 2015
> > - Evaluation dataset: KITTI 2015
> >
> > |  | Data Size | KITTI 2015 (AEPE / F1) |
> > | --- | --- | --- |
> > | RF-Ktrain | 4k | 2.18 / 8.65 |
> > | RF-Ktrain → DFlow-K | 4k → 0.1k | 1.83 / 6.44 |
> >
> > For the above fusion experiment, we use the RF-Ktrain dataset proposed by RealFlow as our base dataset and the KITTI 2015 dataset as the target dataset, respectively. Thereby, we generate a new DFlow dataset, DFlow-K, to capture distinct features of KITTI 2015, which might not be covered by the RF-Ktrain dataset even using KITTI 2015’s texture and motion by the RealFlow approach. Then, we fine-tune the RAFT base network pre-trained on RF-Ktrain with the DFlow-K dataset.
> >
> > When using DFlow-K as a fine-tuning dataset, we achieve favorable performance improvement against the network trained on the RF-Ktrain dataset [RealFlow], which uses the KITTI 2015 training set as a source. We postulate that this improvement is obtained by our method capturing the features of the target dataset distinctive from the given base dataset.
> > We believe that our method can also be fused with other dataset generation method (FlyingChairs, RealFlow, etc.) together as shown above.

---

### Author Response · Authors · 2022-11-19
**Revision Summary**

$\textcolor{#F1433F}{Reviewer \ uY3h}$, $\textcolor{#E7B500}{Reviewer \ IPLs}$, $\textcolor{#A9CF54}{Reviewer \ tPfm}$, $\textcolor{#70B7BA}{Reviewer \ dkLC}$

We thank the reviewers for their constructive review and time. We are pleased that the reviewers recognized our idea of the efficient differentiable data generation pipeline ($\textcolor{#F1433F}{uY3h}$, $\textcolor{#E7B500}{IPLs}$, $\textcolor{#A9CF54}{tPfm}$, $\textcolor{#70B7BA}{dkLC}$),  and the loss function ($\textcolor{#F1433F}{uY3h}$) for comparing dataset-to-dataset. We are glad to hear good quality ($\textcolor{#E7B500}{IPLs}$) and transparency ($\textcolor{#A9CF54}{tPfm}$) of our work.

We summarize the revision as follows:

- Clarifying “SOTA” performance not to confuse → Section abstract and page 7 ($\textcolor{#F1433F}{uY3h}$, $\textcolor{#70B7BA}{dkLC}$)
- Adding discussion and limitations of our work → Section 5 on page 9 ($\textcolor{#F1433F}{uY3h}$, $\textcolor{#E7B500}{IPLs}$)
- Cited and discussed RealFlow, our concurrent work → Section 2 on page 2 ($\textcolor{#F1433F}{uY3h}$)
- Adding summarization to make understanding easy and revising the explanation technical part to be more readable → Section 3 on page 5 and 6 ($\textcolor{#A9CF54}{tPfm}$)
- Adding pseudo-code →  Algorithm 1 on page 6 and Algorithm 2 on page 15 ($\textcolor{#A9CF54}{tPfm}$)
- Comparison between alpha blending and softmax splatting → Table 5 on page 8 ($\textcolor{#A9CF54}{tPfm}$)
- Adding the experiments with varying numbers of data samples → Table 12 on page 20 ($\textcolor{#70B7BA}{dkLC}$)
- Reproducibility concerns → We will release the code and dataset if accepted. ($\textcolor{#F1433F}{uY3h}$, $\textcolor{#A9CF54}{tPfm}$, $\textcolor{#70B7BA}{dkLC}$)

We think that our submission has become more convincing thanks to the invaluable feedback. We again thank the reviewers for their time and comments.

---

### Decision · Program_Chairs · 2023-01-20

**Decision:**

Accept: poster

**Justification For Why Not Higher Score:**

The idea of automated optical flow dataset generation is not new, so the paper is somewhat incremental.

**Justification For Why Not Lower Score:**

Overall, the proposed method is interesting and new, and the paper shows that it performs well via extensive experiments.

**Metareview: Summary, Strengths And Weaknesses:**

The paper proposes an approach for automatically generating datasets for training optical flow estimation models, but optimizing a differentiable data generation pipeline to make examples that are difficult for an optical flow model trained on a generic dataset and easy for a model trained on a target dataset. Models trained on thus generated datasets turn out to perform well in many settings.

The opinions of the reviewers about the paper are mixed, even after the authors' rebuttals. The key points are as follows:

Pros:
1. A new and conceptually simple approach
2. Good performance compared to relevant baselines, demonstrated via extensive experiments
3. Ablation study
4. Computational efficiency compared to AutoFlow.

Cons:
1. Misleading SOTA claims
2. Generated datasets end up being target-domain-specific
3. Limited discussion of limitations
4. Some unclarities in writing
5. No comparison to RealFlow

The authors did a good job of addressing the cons in the rebuttal, so (1), (3), (4), (5)  are largely fixed.

Overall, the proposed method is interesting and new, and the paper shows that it performs well via extensive experiments. Therefore, I recommend acceptance. However, I urge the authors to make sure to take the reviewers' comments into account in the final version.

**Note From Pc:**

if the above contains the word "oral" or "spotlight" please see: "oral" presentation means -> notable-top-5% and "spotlight" means -> notable-top-25%. As stated in our emails, we are disassociating presentation type from AC recommendations